# An electrodiffusive, ion conserving Pinsky-Rinzel model with homeostatic mechanisms

**Marte J. Sætra**[1,2], **Gaute T. Einevoll**[1,2,3], **Geir Halnes**[1,3]*

**1** Centre for Integrative Neuroplasticity, University of Oslo, Oslo, Norway, **2** Department of Physics, University of Oslo, Oslo, Norway, **3** Faculty of Science and Technolgy, Norwegian University of Life Sciences, Ås, Norway

* geir.halnes@nmbu.no

**Data Availability Statement:** All relevant data are within the manuscript, and the computer code is available in online repositories: https://github.com/CINPLA/EDPRmodel and https://github.com/CINPLA/EDPRmodel_analysis.

## Abstract

In most neuronal models, ion concentrations are assumed to be constant, and effects of concentration variations on ionic reversal potentials, or of ionic diffusion on electrical potentials are not accounted for. Here, we present the electrodiffusive Pinsky-Rinzel (edPR) model, which we believe is the first multicompartmental neuron model that accounts for electrodiffusive ion concentration dynamics in a way that ensures a biophysically consistent relationship between ion concentrations, electrical charge, and electrical potentials in both the intra- and extracellular space. The edPR model is an expanded version of the two-compartment Pinsky-Rinzel (PR) model of a hippocampal CA3 neuron. Unlike the PR model, the edPR model includes homeostatic mechanisms and ion-specific leakage currents, and keeps track of all ion concentrations ($Na^+$, $K^+$, $Ca^{2+}$, and $Cl^-$), electrical potentials, and electrical conductivities in the intra- and extracellular space. The edPR model reproduces the membrane potential dynamics of the PR model for moderate firing activity. For higher activity levels, or when homeostatic mechanisms are impaired, the homeostatic mechanisms fail in maintaining ion concentrations close to baseline, and the edPR model diverges from the PR model as it accounts for effects of concentration changes on neuronal firing. We envision that the edPR model will be useful for the field in three main ways. Firstly, as it relaxes commonly made modeling assumptions, the edPR model can be used to test the validity of these assumptions under various firing conditions, as we show here for a few selected cases. Secondly, the edPR model should supplement the PR model when simulating scenarios where ion concentrations are expected to vary over time. Thirdly, being applicable to conditions with failed homeostasis, the edPR model opens up for simulating a range of pathological conditions, such as spreading depression or epilepsy.

## Author summary

Neurons generate their electrical signals by letting ions pass through their membranes. Despite this fact, most models of neurons apply the simplifying assumption that ion concentrations remain effectively constant during neural activity. This assumption is often quite good, as neurons contain a set of homeostatic mechanisms that make sure that ion

**Funding:** This work was funded by the Research Council of Norway (https://www.forskningsradet.no) via the BIOTEK2021 Digital Life project'DigiBrain', grant no 248828 (received by GTE). The funders had no role in study design, data collection and analysis, decision to publish, or preparation of the manuscript.

**Competing interests:** The authors have declared that no competing interests exist.

concentrations vary quite little under normal circumstances. However, under some conditions, these mechanisms can fail, and ion concentrations can vary quite dramatically. Standard models are thus not able to simulate such conditions. Here, we present what to our knowledge is the first multicompartmental neuron model that accounts for ion concentration variations in a way that ensures complete and consistent ion concentration and charge conservation. In this work, we use the model to explore under which activity conditions the ion concentration variations become important for predicting the neurodynamics. We expect the model to be of great value for the field of neuroscience, as it can be used to simulate a range of pathological conditions, such as spreading depression or epilepsy, which are associated with large changes in extracellular ion concentrations.

## Introduction

The neuronal action potential (AP) is generated by a transmembrane influx of $Na^+$, which depolarizes the neuron, followed by an efflux of $K^+$, which repolarizes it. Likewise, all neurodynamics is fundamentally about the movement of ions, which are the charge carriers in the brain. Therefore, it might seem peculiar that most models of neuronal activity are based on the approximation that the concentrations of the main charge carriers ($Na^+$, $K^+$, and $Cl^-$) do not change over time. This approximation is, for example, incorporated in the celebrated Hodgkin-Huxley model [1], and a large number of later models based on a Hodgkin-Huxley type formalism (see, e.g., [2–7]).

Setting the ion concentrations to not change over time is often a fairly good approximation. The reason is that the number of ions that need to cross the membrane to charge up the neuron by, say, an AP worth of millivolts, is too small to have any notable impact on ion concentrations on either side of the membrane (see, e.g., Box 2.2 in [8]), meaning that concentration changes on a short time scale can be neglected. On a longer time-scale, the ionic exchange due to APs (or other neuronal events), is normally reversed by a set of homeostatic mechanisms such as ion pumps and cotransporters, which work to maintain constant baseline concentrations. In Hodgkin-Huxley type models, the large number of ion pumps, cotransporters and passive ionic leakages that strive towards maintaining baseline conditions are therefore not explicitly modeled. Instead, they are simply assumed to do their job and are grouped into a single *passive* and non-specific leakage current $I_{leak} = g_{leak}(\phi_m - E_{leak})$, which determines the cell's resting potential (for a critical study of this approximation, see [9]).

Another approximation commonly applied by modelers of neurons is that the extracellular potential is constant and grounded ($\phi_e = 0$) so that the only voltage variable that one needs to worry about when simulating neurodynamics is the transmembrane potential ($\phi_m$). This assumption is implicit in the majority of morphologically explicit models of neurons, where the (spatial) signal propagation in dendrites and axons are computed using the cable equation (see, e.g., [10–12]). Cable-equation based, multicompartmental neuronal models are widely used within the field of neuroscience, both for understanding dendritic integration and neuronal response properties at the single neuron level (see, e.g., [3, 4, 6, 7]) and for exploring the dynamics of large neuronal networks (see e.g., [13–15]). They are even used in the context of performing forward modeling of extracellular potentials, such as local field potentials (LFP), the electrocorticogram (ECoG), and electroencephalogram (EEG) (see, e.g., [16–18]), despite the evident inconsistency involved when first computing neurodynamics under the approximation that $\phi_e = 0$ (Fig 1A), and then in the next step using this dynamics to predict a nonzero $\phi_e$ (Fig 1B). The approximation is nevertheless useful since $\phi_e$ is typically so much smaller than

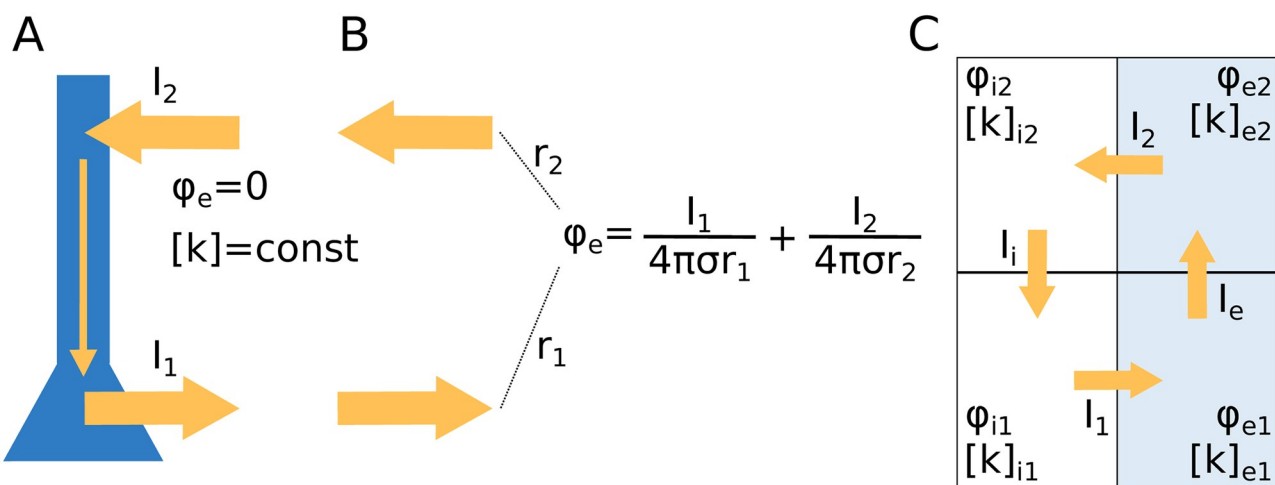

**Fig 1. Modeling intra- and extracellular dynamics: Standard theory vs. unified framework.** **(A)** The dynamics of the membrane potential ($\phi_m$) and transmembrane currents of neurons are typically modeled using cable theory. It is then assumed that the extracellular environment is grounded ($\phi_e = 0$). Typically, it is also assumed that ion concentrations both in the intra- and extracellular space are constant, so that also ionic reversal potentials remain constant. **(B)** When knowing the transmembrane neuronal currents (as computed in **(A)**), standard volume conductor theory [20, 21] allows us to estimate the extracellular potential, which is computed as the sum of neuronal point-current sources weighted by their distance to the recording location. An underlying assumption is that fluctuations in $\phi_e$ (as computed in **(B)**) are so small that they have no effect on the neurodynamics (as computed in **(A)**), i.e., there is no ephaptic coupling. Another underlying assumption (cf. constant ion concentrations) is that extracellular diffusive currents do not affect electrical potentials. **(C)** We propose a unified, electrodiffusive framework for intra- and extracellular ion concentration and voltage dynamics, assuring a consistent relationship between ion concentrations, electrical charge, and electrical potential in all compartments.

$\phi_m$ that the (ephaptic) effect of $\phi_e$ on neurodynamics can be neglected without severe loss in accuracy [19].

There are, however, scenarios where the assumptions of constant ion concentrations and a grounded extracellular space are not justifiable. Notably, large-scale extracellular ion concentration changes are a trademark of several pathological conditions, including epilepsy and spreading depression [22–25]. In these cases, neurons are unable to maintain their baseline conditions because they for various reasons are too active and/or their homeostatic mechanisms are too slow. During spreading depression, the extracellular $K^+$ concentration can change from a baseline value of about 3-5 mM to pathological levels of several tens of mM, and the increased $K^+$ concentration tends to coincide with a slow, direct-current (DC) like drop in the extracellular potential, which may be several tens of millivolts in amplitude [25, 26], and can give rise to large spatial gradients. For example, one experiment saw the extracellular $K^+$-concentration and $\phi_e$ vary by as much as 30 mM and 20 mV, respectively, over the hippocampal depth [26]. Such dramatic gradients in the extracellular environment are likely to have a strong impact on the dynamical properties of neurons, both through the concentration-dependent changes in ion-channel reversal potentials [27–29] and putatively through a direct ephaptic effect from $\phi_e$ on the membrane potential.

The construction of accurate neuron models that include ion concentration dynamics (and conservation) poses two key challenges. Firstly, ion conserving models need a finely adjusted balance between the homeostatic machinery and all passive and active ion-specific currents so that all ion concentrations, as well as voltages, vary in a biophysically realistic way over time when the neuron is active. Secondly, in spatially extended models, ions will not move only across membranes, but also within the extracellular and intracellular space. Such ionic movement may be propelled both by diffusion and electrical drift. Ionic diffusion can, in principle, affect the electrical potential (since ions carry charge), and the electrical potential can, in principle, affect ion concentration dynamics (since ions drift along potential gradients) [30–32].

Accurate modeling of such systems thus requires a unified, electrodiffusive framework that ensures a physically consistent relationship between ion concentrations, charge density, and electrical potentials.

Intra- or extracellular electrodiffusion is not an issue in single-compartment models, of which there are quite a few that incorporate ion concentration dynamics in a more or less consistent way [28, 29, 33–47]. Single compartment models are useful in many aspects. However, in order to represent morphological features of neurons, such as e.g., differential expression of ion channels in the soma versus dendrites, or account for transport processes in the space inside or outside neurons, one needs models with more than a single compartment. Among the several morphologically explicit models that have included homeostatic machinery and explicitly simulated ion concentration dynamics (see e.g., [27, 48–57]), neither have accounted for the electrodiffusive coupling between the movement of ions and electrical potentials (see Results section titled Loss in accuracy when neglecting electrodiffusive effects on concentration dynamics). Hence, to our knowledge, no morphologically explicit neuron model has so far been developed that ensures biophysically consistent dynamics in ion concentrations and electrical potentials during long-time activity, although useful mathematical framework for constructing such models are available [58–62].

The goal of this work is to propose what we may refer to as "a minimal neuronal model that has it all". By "has it all", we mean that it (1) has a spatial extension, (2) considers both extracellular- and intracellular dynamics, (3) keeps track of all ion concentrations (Na$^+$, K$^+$, Ca$^{2+}$, and Cl$^-$) in all compartments, (4) keeps track of all electrical potentials ($\phi_m$, $\phi_e$, and $\phi_i$—the latter being the intracellular potential) in all compartments, (5) has differential expression of ion channels in soma versus dendrites, and can fire somatic APs and dendritic calcium spikes, (6) contains the homeostatic machinery that ensures that it maintains a realistic dynamics in $\phi_m$ and all ion concentrations during long-time activity, and (7) accounts for transmembrane, intracellular and extracellular ionic movements due to both diffusion and electrical migration, and thus ensures a consistent relationship between ion concentrations and electrical charge. Being based on a unified framework for intra- and extracellular dynamics (Fig 1C), the model thus accounts for possible ephaptic effects from extracellular dynamics, as neglected in standard feedforward models based on volume conductor theory (Fig 1A and 1B). By "minimal" we simply mean that we reduce the number of spatial compartments to the minimal, which in this case is four, i.e., two neuronal compartments (a soma and a dendrite), plus two extracellular compartments (outside soma and outside dendrite). Technically, the model was constructed by adding homeostatic mechanisms and ion concentration dynamics to an existing model, i.e., the two-compartment Pinsky-Rinzel (PR) model [3], and embedding in it a consistent electrodiffusive framework, i.e., the previously developed Kirchhoff-Nernst-Planck framework [31, 32, 60, 62]. For the remainder of this paper, we refer to our model as the electrodiffusive Pinsky-Rinzel (edPR) model.

The remainder of this article is organized as follows. First, we present the edPR model and illustrate the numerous variables that it can simulate. Next, we show that the edPR model can reproduce the firing properties of the original PR model. By running long-time simulations (several minutes of biological time) on both models, we identify the firing conditions under which the two models maintained a similar firing pattern, and under which conditions concentration effects became important so that dynamics of the edPR model diverged from the original PR model over time. Finally, we use the edPR model to explore the validity of some important assumptions commonly made in the field of computational neuroscience, regarding the decoupling of electrical and diffusive signals. We believe that the edPR model will be of great value for the field of neuroscience, partly because it gives a deepened insight into the balance between neuronal firing and ion homeostasis, partly because it lends itself to explore

under which conditions the common modeling assumption of constant ion concentrations is warranted, and most importantly because it opens for more detailed mechanistic studies of pathological conditions associated with large changes in ion concentrations, such as epilepsy and spreading depression [22–25].

## Results

### An electrodiffusive Pinsky-Rinzel model

The here proposed electrodiffusive Pinsky-Rinzel (edPR) model is inspired by the original Pinsky-Rinzel (PR) model [3], which is a two-compartment (soma + dendrite) version of a CA3 hippocampal cell model, initially developed by Traub et al. [2]. In the original PR model, the somatic compartment contains $Na^+$, and $K^+$ delayed rectifier currents ($I_{Na}$ and $I_{K–DR}$), while the dendritic compartment contains a voltage-dependent $Ca^{2+}$ current ($I_{Ca}$), a voltage-dependent $K^+$ afterhyperpolarization current ($I_{K–AHP}$), and a $Ca^{2+}$-dependent $K^+$ current ($I_{K–C}$). In addition, both compartments contain passive leakage currents. Despite its small number of compartments and conductances, the PR model can reproduce a variety of realistic firing patterns when responding to somatic or dendritic stimuli, including somatic APs and dendritic calcium spikes.

In the edPR model, we have adopted all mechanisms from the original PR model. In addition, we have (i) made all ion channels and passive leakage currents ion-specific, (ii) included $3Na^+/2K^+$ pumps ($I_{pump}$), $K^+/Cl^-$ cotransporters ($I_{KCC2}$), $Na^+/K^+/2Cl^-$ cotransporters ($I_{NKCC1}$), and $Ca^{2+}/2Na^+$ exchangers ($I_{Ca–dec}$), and (iii) included two extracellular compartments (outside soma + outside dendrite). To compute the dynamics of the edPR, we used an electrodiffusive KNP-framework for consistently computing the voltage- and ion concentration dynamics in the intra- and extracellular compartments [60]. The model is summarized in Fig 2 and described in details in the Methods section.

### Key dynamical variables in the electrodiffusive Pinsky-Rinzel model

While the key variable in the original PR model is the membrane potential $\phi_m$, the edPR model allows us to compute a multitude of variables relevant to neurodynamics. The functionality of the edPR model is illustrated in Fig 3, which shows a 60 s simulation where the model fires at 1 Hz for 10 s. We have plotted a selection of output variables, including the membrane potentials (Fig 3A and 3B), extracellular potentials (Fig 3C and 3D), the dynamics of all ion concentrations in all compartments (Fig 3E–3H), concentration effects on ionic reversal potentials (Fig 3I–3J), concentration effects on the electrical conductivity of the intra- and extracellular medium (Fig 3K), and ATP consumption (Fig 3L) of the $3Na^+/2K^+$ pumps and $Ca^{2+}/2Na^+$ exchangers.

Unlike neuronal models based on cable theory, where $\phi_e$ is assumed to be zero so that $\phi_m = \phi_i$, the edPR model computes $\phi_m$, $\phi_i$, and $\phi_e$ from a consistent framework where ephaptic effects from $\phi_e$ on $\phi_m$ are accounted for (Fig 3C). Due to the electrical coupling between the soma and dendrite, the fluctuations in $\phi_m$ were similar in these compartments, and a more detailed analysis of the AP shapes is found further below. While an action potential essentially gave a depolarization followed by a repolarization of $\phi_m$, its extracellular signature was essentially a voltage drop (to about -5 mV) followed by a voltage increase (to about +5 mV). This biphasic response of the extracellular AP signature has been seen in several studies (for an analysis, see [20, 21]). In experimental recordings, amplitudes in $\phi_e$ fluctuations are typically on the order of 100 $\mu$V, which is much smaller than that predicted by the edPR model. The discrepancy is an artifact that is mainly due to the 1D approximation in the edPR model (see

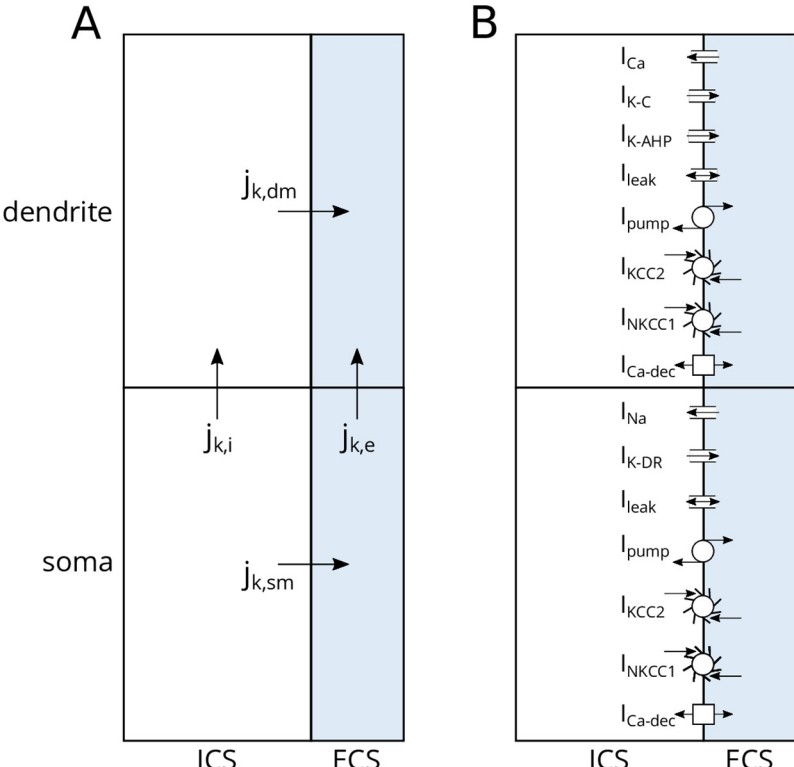

**Fig 2. edPR model architecture. (A)** Two plus two compartments (soma + dendrite), with intracellular space to the left and extracellular space to the right. Two kinds of fluxes of different ion species k are involved: transmembrane fluxes ($j_{k,dm}$, $j_{k,sm}$) and intra- and extracellular fluxes ($j_{k,i}$, $j_{k,e}$). The dynamics of the potential $\phi$ and ion concentration dynamics in all compartments were computed using an electrodiffusive framework, ensuring bulk electroneutrality and a consistent relationship between ion concentrations, electrical charge, and voltages. **(B)** Active currents were taken from the original PR model [3]. In the soma, these consisted of Na$^+$ and K$^+$ delayed rectifier currents ($I_{Na}$ and $I_{K-DR}$). In the dendrite, these consisted of a voltage-dependent Ca$^{2+}$ current ($I_{Ca}$), a Ca$^{2+}$-dependent K$^+$ current ($I_{K-C}$), and a voltage-dependent K$^+$ afterhyperpolarization current ($I_{K-AHP}$). Ion specific passive (leakage-) currents and homeostatic mechanisms were taken from a previous model by Wei et al. [45], and were identical in the soma and dendrite. These included Na$^+$, K$^+$ and Cl$^-$ leak currents, a 3Na$^+$/2K$^+$ pump ($I_{pump}$), a K$^+$/Cl$^-$ cotransporter ($I_{KCC2}$), and a Na$^+$/K$^+$/2Cl$^-$ cotransporter ($I_{NKCC1}$). In addition, the soma and dendrite included a Ca$^{2+}$/2Na$^+$ exchanger ($I_{Ca-dec}$), providing an intracellular Ca$^{2+}$ decay similar to that in the PR model.

Discussion). The dendritic extracellular potential (Fig 3D) was by definition zero at all times, as this compartment was used as the reference point for the potential.

The effect of neuronal firing on the ion concentration dynamics is illustrated in Fig 3E–3H. Before the stimulus onset, the cell was resting at approximately -68 mV, and ion concentrations remained at baseline values. During AP firing, the ion concentrations varied in a jigsaw-like fashion in all compartments, except for Ca$^{2+}$, which returned to baseline between each AP and showed notable variation only inside/outside the dendrite since the soma contained no Ca$^{2+}$ channels. As the extracellular volume was set to be half as big as the intracellular volume, changes in extracellular ion concentrations were about twice as big as the changes in intracellular ion concentrations. The jigsaw pattern was most pronounced for the K$^+$ and Na$^+$ concentrations, as these were the main mediators of the APs (Fig 3E–3H). The pattern reflects a cycle of (i) incremental steps away from baseline concentrations, which were mediated by the complex of mechanisms active during the APs, followed by (ii) slower decays back towards baseline, which were mediated by pumps and cotransporters working between the APs. In this simulation, the decay was incomplete, so that concentrations reached gradually larger peak

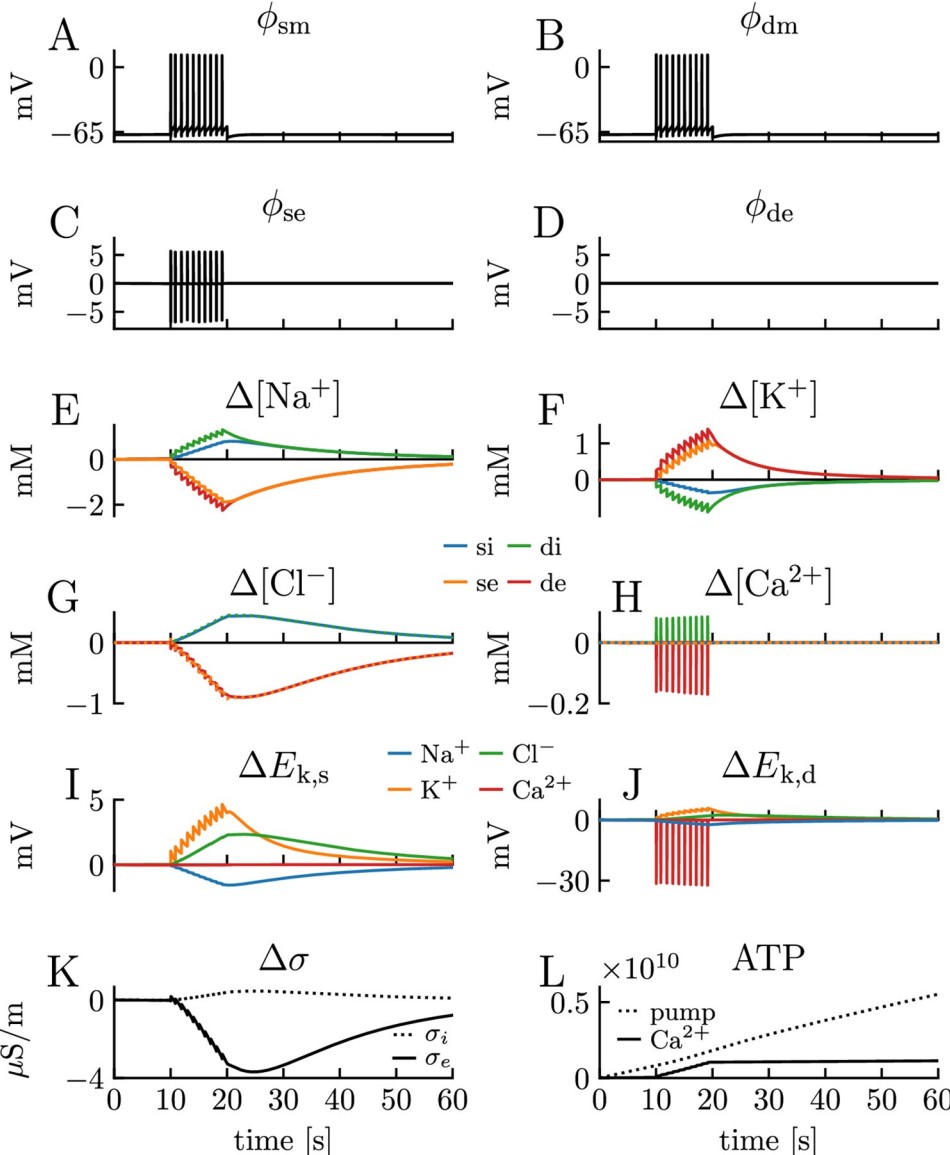

**Fig 3. Output of the edPR model.** A 27 pA step-current injection was applied to the somatic compartment between $t = 10$ s and $t = 20$ s, and the model responded with a firing rate of 1 Hz. **(A-B)** The membrane potential $\phi_m$ of the soma and the dendrite, respectively. **(C-D)** The extracellular (index e) potential $\phi_e$ of the soma (index s) and the dendrite (index d), respectively. The dendritic extracellular compartment was chosen as the reference point when calculating potentials, so $\phi_{de}$ was zero by definition. Since amplitudes in $\phi_m$ were so much larger than for $\phi_e$, intracellular (index i) potentials ($\phi_i = \phi_e + \phi_m$) were similar to $\phi_m$, and therefore not shown. **(E-H)** Ion concentrations dynamics of all ion species k ($Na^+$, $Cl^-$, $K^+$, $Ca^{2+}$) in all four compartments shown in terms of their deviance from baseline concentrations. **(I-J)** Changes in reversal potentials for all ion species in the soma and the dendrite, respectively. **(K)** Change in conductivity of the intra- and extracellular media ($\sigma_i$ and $\sigma_e$, respectively). **(L)** Accumulative number of ATP molecules consumed by the $3Na^+$/$2K^+$ pumps and $Ca^{2+}$/$2Na^+$ exchangers.

values by each consecutive AP. However, as we show later (see Section titled The edPR model predicts homeostatic failure due to high firing rate), the concentrations did, in this case, approach a firing-frequency dependent steady state.

When the firing ceased in Fig 3, the pumps and cotransporters could work uninterruptedly to re-establish the baseline ion concentrations. The resting membrane potential of about -68

mV, was recovered quite rapidly (ms timescale). After this, the slower recovery process of the ion concentration was due to an electroneutral exchange of ions between the neuron and the extracellular space. A full recovery of the baseline concentrations took on the order of 80 s (confirmed by running a longer simulation than the one shown in Fig 3).

As ion concentrations varied during the simulation, so did the ionic reversal potentials, $E_k$(Fig 3I–3J). The by far largest change was seen for the $Ca^{2+}$ reversal potential in the dendrite ($E_{k,d}$), which dropped by as much as -30 mV during an AP, (i.e., from a baseline value of 124 mV to 94 mV). The explanation is that the basal intracellular $Ca^{2+}$-concentration is extremely low (100 nM) compared to the concentrations of other ion species (several mM), and therefore experienced a much larger relative change during the simulation. Among the main charge carriers ($Na^+$, $Cl^-$, $K^+$), the lowest concentration is found for $K^+$ in the extracellular space (Table 5 in Methods). For that reason, the second largest change in reversal potential was found for $E_K$, which increased by about 5 mV (i.e., from a basal value of -84 mV to -79 mV) in both the soma and dendrite. The changes in $E_{Ca}$ and $E_K$ had a relatively minor impact on the firing pattern in the shown simulations, as the relative change in the driving force $\phi_m - E_k$ was not that severe.

The conductivities ($\sigma$) of the intra- and extracellular bulk solutions depend on the availability of free charge carriers, and are in the edPR model functions of the ion concentrations and their mobility (cf. Eq 19). The changes in $\sigma$ were minimal during the conditions simulated here (Fig 3K), i.e., $\sigma$ varied by a few $\mu$S/m over the course of the simulation, while the basal levels were approximately 0.11 S/m and 0.59 S/m for the intra- and extracellular solutions, respectively.

Finally, the $3Na^+/2K^+$ pump and $Ca^{2+}/2Na^+$ exchanger use energy in the form of ATP to move ions against their gradients. The $3Na^+/2K^+$ pump exchanges 3 $Na^+$ ions for 2 $K^+$ ions, and consumes one ATP per cycle [63], while we assumed that the $Ca^{2+}/2Na^+$ exchanger consumed 1 ATP per cycle (i.e., per $Ca^{2+}$ exchanged, as in [64]). As the edPR model explicitly models these processes, we could compute the ATP (energy) consumption of the pumps during the simulation. Fig 3L shows the accumulative number of ATP consumed from the onset of the simulation. The $3Na^+/2K^+$ pump was constantly active, as it combated leakage currents and worked to maintain the baseline concentration even before stimulus onset. Before stimulus onset, it consumed ATP at a constant rate (linear curve), which increased only slightly at $t = 10$ s when the neuron started to fire (small dent in the curve). As the neuron did not contain any passive leakage of $Ca^{2+}$, the $Ca^{2+}/2Na^+$ exchangers were only active while the neuron was firing. During firing, the $Ca^{2+}/2Na^+$ exchanger combated the $Ca^{2+}$ entering through the dendritic $Ca^{2+}$ channels, and then consumed approximately the same amount of energy as the $3Na^+/2K^+$ pump (parallel curves). A high metabolic cost of dendritic $Ca^{2+}$ spikes has previously been reported also in cortical layer 5 pyramidal neurons [64].

We note that the edPR model had a stable resting state before stimulus onset and that it returned to this resting state after the stimulus had been turned off. In this resting state, ion concentrations remained constant, and $\phi_m$ was approximately -68 mV. This resting equilibrium was due to a balance between the ion-specific leakage channels, pumps, and cotransporters, which we adopted from previous studies (see Methods). However, the existence of a homeostatic equilibrium was not highly sensitive to the choice of model parameters. As we confirmed through a sensitivity analysis, varying membrane parameters (one by one) with ±15% of their default values did not abolish the existence of a stable resting state, but shifted the resting potential by maximally ±3 mV (Fig 4A) and the resting concentrations by maximally 5% (Fig 4B–4E).

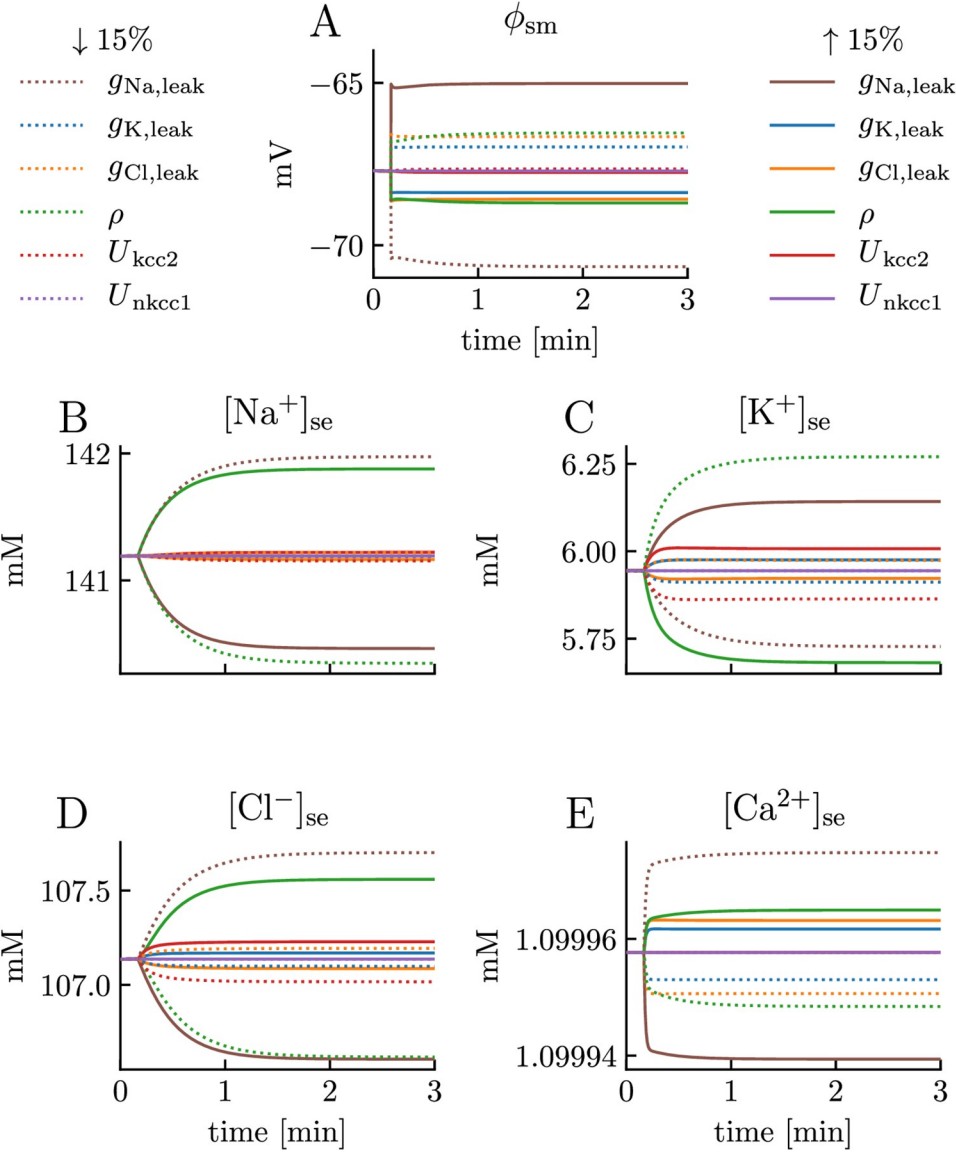

**Fig 4. Sensitivity analysis.** Sensitivity of **(A)** the somatic membrane potential ($\phi_{sm}$) and **(B-E)** ion concentrations outside the soma to variations of the leak conductances $\overline{g}_{Na,leak}$, $\overline{g}_{K,leak}$, and $\overline{g}_{Cl,leak}$, the ATPase pump strength $\rho$, and the co-transporter strengths $U_{nkcc1}$ and $U_{kcc2}$. The model was run for 10 s with default parameters. At $t$ = 10 s, selected parameters were changed, one per simulation, by ±15% of their default value. In all cases, the model approached a new steady state during the 3 min simulation, which was not dramatically different from the default steady state. The resting potential was most sensitive to $\overline{g}_{Na,leak}$. This was not surprising, as Na$^+$ has the reversal potential (57 mV) that is furthest away from the resting potential ($\approx$ -68 mV), making the driving force ($\phi_m - E_k$) largest for Na$^+$. All concentration variables were most sensitive either to $\overline{g}_{Na,leak}$ or $\rho$. For [Ca$^{2+}$]$_{se}$ and [Cl$^-$]$_{se}$ the sensitivity to these parameters were indirect, i.e., through their effects on the resting potential and driving forces. **(A-E)** Results only shown for somatic compartments, as they were almost identical in the the dendritic compartments. Only extracellular concentrations were shown, but intracellular concentrations followed the same time coarse and intracellular deviances from default values were smaller (due to larger intracellular volume fraction). As we showed in Fig 3L, the Ca$^{2+}$/2Na$^+$ exchanger is not active during rest, and it was therefore not included in the sensitivity analysis.

## The edPR model reproduces the short term firing properties of the original PR model

A motivation behind basing the electrodiffusive (edPR) model on a previously developed (PR) model, was that we wanted to use the firing properties of the original PR model as a "ground truth" when constraining the edPR model. In particular, we wanted the edPR model to qualitatively reproduce the interplay between somatic action potentials and dendritic $Ca^{2+}$ spikes, as this was an essential feature of the original PR model [3]. In the PR model, this interplay depended strongly on the coupling strength (coupling conductance) between the soma and dendrite compartment. A weak coupling resulted in a wobbly ping-pong effect, where a somatic AP triggered a dendritic $Ca^{2+}$ spike, which in turn fed back to the soma, giving rise to secondary oscillations in $\phi_m$ (Fig 5A). With a strong (about five times stronger) coupling, the somatic and dendritic responses became more similar in shape, as expected (Fig 5B).

Since the edPR model contained membrane mechanisms and ephaptic effects not present in the PR model, selected parameters in the edPR model had to be re-tuned in order to obtain similar firing as the PR model (see Methods). With the selected parameterization of the edPR model (see the Parameterizations section), we were able to reproduce the characteristic features seen in the PR model for a weak (Fig 5C) and strong (about five times stronger) coupling between the soma and dendrite (Fig 5D).

## The edPR model predicts homeostatic failure due to high firing rate

As previously discussed, the PR model was, as most existing neuronal models, constructed under the assumption that ion concentration effects are negligible, an assumption that is justified for short term neurodynamics, and for long term dynamics provided that the activity level is sufficiently low for the homeostatic mechanisms to maintain concentrations close to baseline over time. Hence, we expect there to be a scenario (S1) with a moderately low firing rate, where the PR and edPR models can fire continuously and regularly over a long time exhibiting similar firing properties, and another scenario (S2) with a higher firing rate, where the PR and edPR models exhibit similar firing properties initially in the simulation, but where the dynamics of the two models diverge over time due to homeostatic failure accounted for by the edPR model, but not the PR model (which ad hoc assume perfect homeostasis). Simulations of two such scenarios are shown in Figs 6 and 7.

To simulate scenario S1, the PR model (Fig 6A and 6B) and edPR model (Fig 6C–6J) were given a constant input (see figure caption) that gave them a firing rate of 1 Hz. Both models settled at a regular firing rate, and in neither of them the firing pattern changed over time, even in simulations of as much as an hour of biological time. For the edPR model, the S1 scenario is the same as that simulated for only a brief period in Fig 3. There, we observed that the ion concentrations gradually changed during the first seconds after the onset of stimulus (Fig 3E–3H). However, for endured firing, the ion concentrations and reversal potentials settled on a (new) dynamic steady state (Fig 6E–6J), where they deviated by $\sim$1-5 mM from the baseline concentrations during rest (i.e., for edPR receiving no input). The apparent "thickness" of the curves (e.g., the red curve for $K^+$ in Fig 6H) is due to concentration fluctuations at the short time scale of AP firing. However, after each AP, the homeostatic mechanisms managed to re-establish ionic gradients before the next AP occurred, so that no slow concentration-dependent effect developed in the edPR model at a long time scale.

To simulate scenario S2, the PR model (Fig 7A and 7B) and edPR model (Fig 7C–7J) were given a constant input (see figure caption) that gave them a firing rate of about 3 Hz. The PR model, which included no concentration-dependent effects, settled on a regular firing rate that it could maintain for an arbitrarily long time. Unlike the PR model, the edPR model did not

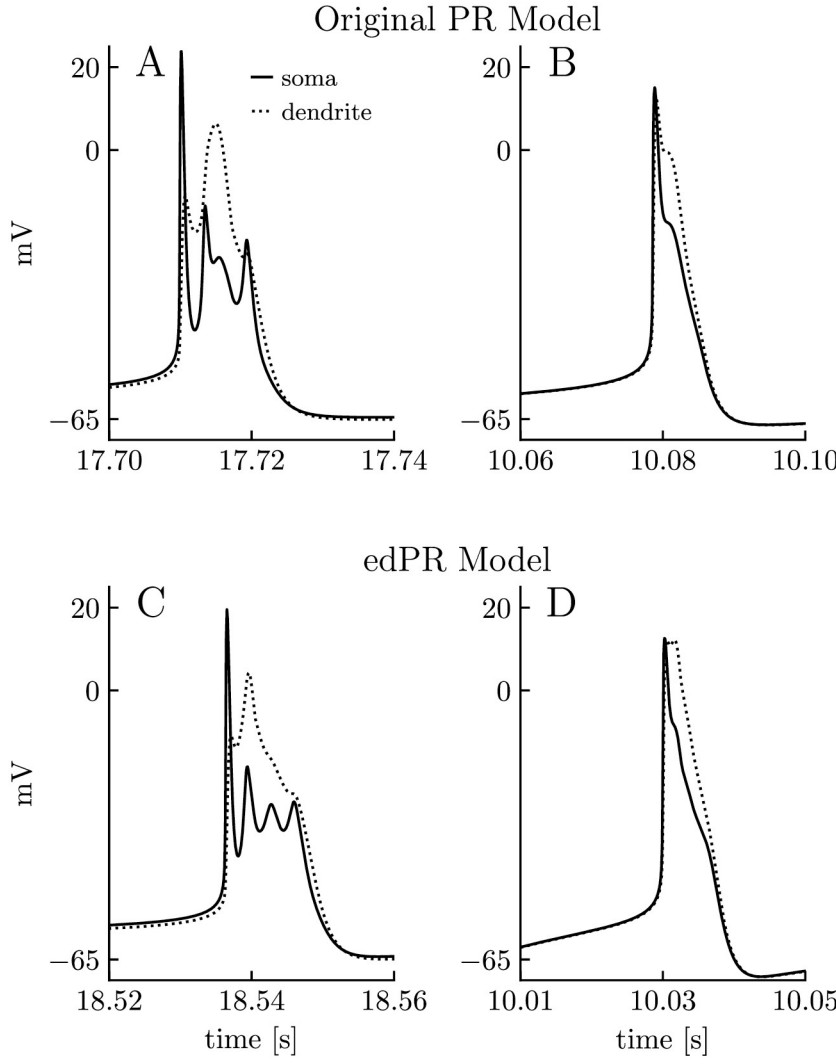

**Fig 5. Short term dynamics of the PR and edPR models.** The original PR model (top row) and the edPR model (bottom row) exhibit the same spike shape characteristics. **(A)** Spike shape in PR model for weak coupling (low coupling conductance) between the soma and the dendrite. **(B)** Spike shape in PR model for strong coupling (high intracellular conductivity) between the soma and the dendrite. **(C)** Spike shape in edPR model for weak coupling between the soma and the dendrite. **(D)** Spike shape in edPR model for strong coupling between the soma and the dendrite. **(A-D)** A step-stimulus current was turned on at $t$ = 10 s, with stimulus strength being 1.35 $\mu$A/cm$^2$ in **(A)**, 0.78 $\mu$A/cm$^2$ in **(B)**, 31 pA in **(C)**, and 27 pA in **(D)**. The panels show snapshots of a selected spike. See the Parameterizations section in Methods for a full description of the parameters used.

settle at a steady state, but had a firing rate of $\sim$ 3 Hz only for a period of $\sim$ 5 s after stimulus onset. During this period, the ion concentrations gradually diverged from the baseline levels (Fig 7G–7J). The corresponding changes in ionic reversal potentials (Fig 7E and 7F) affected the neuron's firing properties and caused its firing rate to gradually increase before it eventually entered the depolarization block and got stuck at about $\phi_m$ = −30mV. The main explanation behind the altered firing pattern was the change in the K$^+$ reversal potential, which, for example, at 9 s after stimulus onset ($t$ = 19 s) had increased by as much as 20 mV from baseline. This shift led to a depolarization of the neuron, which explains both the (gradually) increased firing rate and the (final) depolarization block, i.e., the condition where $\phi_m$ could no longer repolarize to levels below the firing threshold, and AP firing was abolished due to a permanent

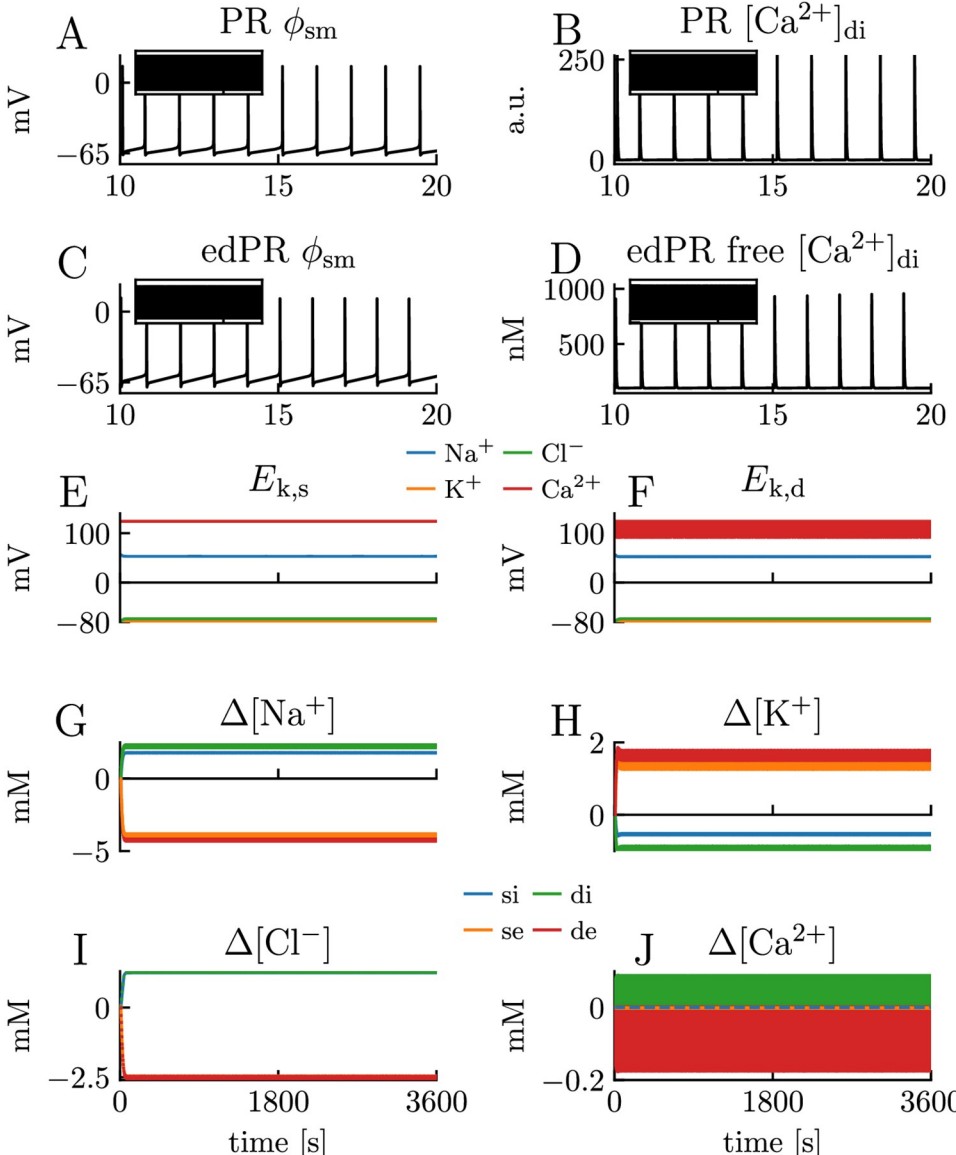

**Fig 6. Model comparison for scenario with low frequency firing.** Simulations on the PR model and edPR model when both models are driven by a constant input, giving them a firing rate of about 1 Hz. Simulations covered one hour (3600 s) of biological time. **(A-D)** A 10 s sample of the dynamics of the somatic membrane potential $\phi_{sm}$ and dendritic (free) $Ca^{2+}$ concentration in the PR model **(A-B)** and edPR model **(C-D)**. This regular firing pattern was sustained over the full 3600 s simulation in both models (inset panels). **(D)** Of the total amount of intracellular $Ca^{2+}$, only 1% (as plotted) was assumed to be free (unbuffered). **(E-F)** Ionic reversal potentials and **(G-J)** ion concentrations in the edPR model did not vary on a long time scale. Indices $i$, $e$, s, and $d$ indicate *intracellular, extracellular, soma*, and *dendrite*, respectively. **(A-J)** Stimulus onset was $t = 10$ s in both models, and stimulus strength was $i_{stim} = 0.78 \mu A/cm^2$ in the PR model **(A-B)** and $i_{stim} = 27$pA in the edPR model **(C-J)**. See the Parameterizations section in Methods for a full description of the parameters used.

inactivation of active $Na^+$ channels. Neuronal depolarization block is a well-studied phenomenon, which is often caused by high extracellular $K^+$ concentrations [65].

The homeostatic failure in S2 was due to the edPR model having a too high firing rate for the ion pumps and cotransporters to maintain ion concentrations close to baseline. The firing rate of 3 Hz was the limiting case (found by trial and error), i.e., for lower firing rates than this, the

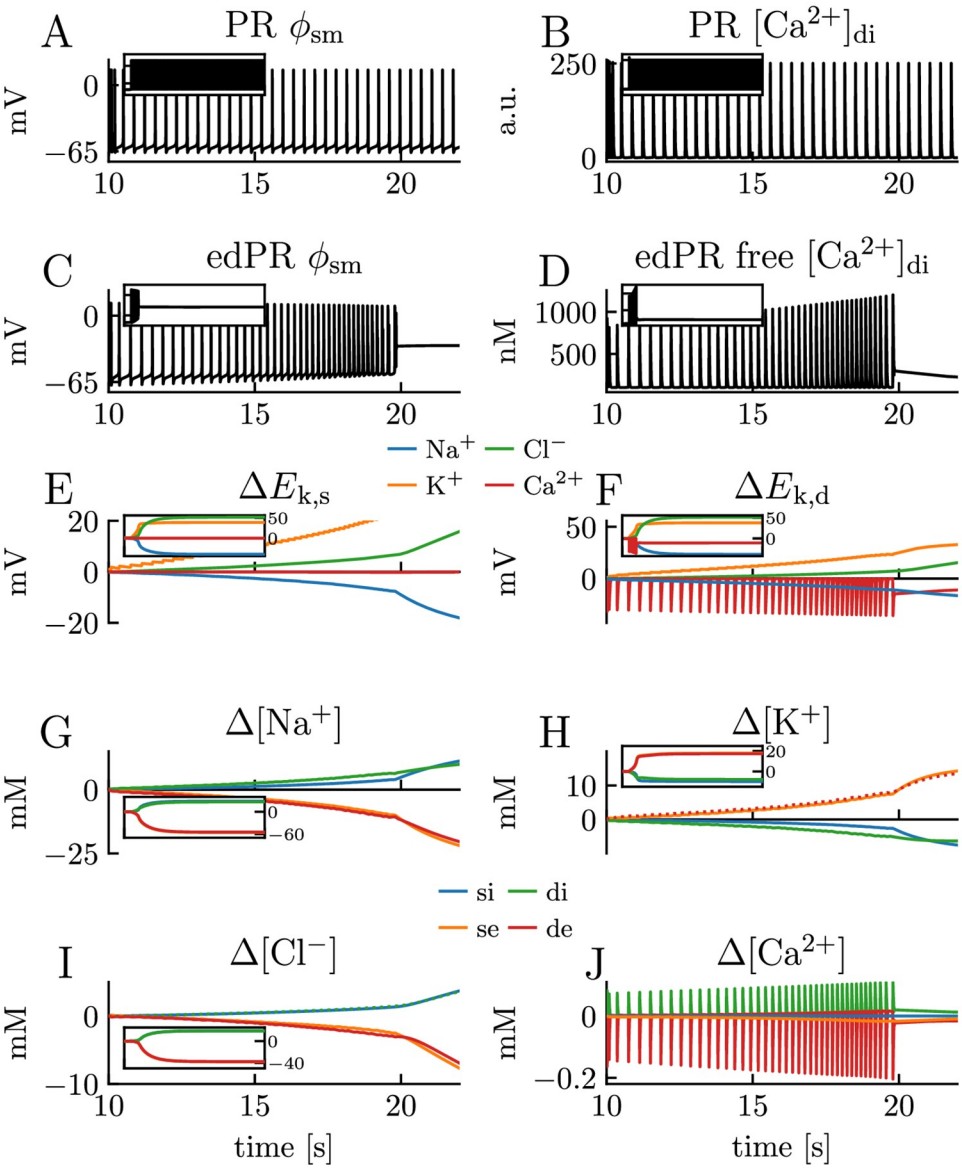

**Fig 7. Model comparison for scenario with high frequency firing.** Simulations on the PR model and edPR model when both models are driven by a constant input, giving them a firing rate of about 3 Hz. Simulations covered 200 s of biological time. **(A-D)** A 12 s sample of the dynamics of the somatic membrane potential $\phi_{sm}$ and dendritic (free) $Ca^{2+}$ concentration in the PR model **(A-B)** and edPR model **(C-D)**. The regular firing pattern in the PR model **(A-B)** was sustained over the full 200 s simulation (inset panels), while the edPR model stopped firing and entered depolarization block around $t = 20$ s. **(D)** Of the total amount of intracellular $Ca^{2+}$, only 1% (as plotted) was assumed to be free (unbuffered). **(E-F)** Ionic reversal potentials and **(G-J)** ion concentrations in the edPR model varied throughout the simulation, and gradually diverged from baseline conditions. Indices $i$, $e$, $s$, and $d$ indicate *intracellular, extracellular, soma*, and *dendrite*, respectively. Main panels show 12 s samples of the ion concentration dynamics, while insets show the dynamics over the full 200 s simulations. **(A-J)** Stimulus onset was $t = 10$ s in both models, and stimulus strength was $i_{stim} = 1.55\mu A/cm^2$ in the PR model **(A-B)** and $i_{stim} = 48$pA in the edPR model **(C-J)**. See the Parameterizations section in Methods for a full description of the parameters used.

model could maintain regular firing for an arbitrarily long time. As many neurons can fire at quite high frequencies, a tolerance level of 3 Hz might seem a bit low, and we here provide some comments to this. Firstly, we note that the edPR model could fire at 3 Hz (and gradually higher frequencies) for about 9 s, and could also maintain a higher firing rate than this for a limited

time. Secondly, the PR model, and thus the edPR model, represented a hippocampal CA3 neuron, which has been found to have an average firing rate of less than 0.5 Hz [66], so that endured firing of $\geq$ 3 Hz may be abnormal for these neurons. Thirdly, under biological conditions, glial cells, and in particular astrocytes, provide additional homeostatic functions [67] that were not accounted for in the edPR model, and the inclusion of such functions would probably increase the tolerance level of the neuron. Fourthly, the (3 Hz) tolerance level was a consequence of modeling choices and could be made higher, e.g., by increasing pump rates or compartment volumes. However, we did not do any model tuning in order to increase the tolerance level, as we, in light of the above arguments, considered a 3 Hz tolerance level to be acceptable.

## The edPR model predicts homeostatic failure due to impaired homeostatic mechanisms

Above we simulated homeostatic failure occurring because the firing rate became too high for the homeostatic mechanisms to keep up (S2). Homeostatic failure may also occur due to impairment of the homeostatic mechanisms, either due to genetic mutations (see, e.g., [68]) or because the energy supply is reduced, such as after a stroke (see, e.g., [25]). Here, we have used the edPR model to simulate a version of this, i.e., a third scenario (S3) where the ATP-dependent mechanisms, that is, the $3Na^+/2K^+$ pumps and the $Ca^{2+}/2Na^+$ exchangers, were turned off.

In S3, the neuron received no external input, so that the dynamics of the neuron was solely due to gradually dissipating transmembrane ion concentration gradients. After an initial transient, we observed a slow and gradual increase in the membrane potential for about 48 s (Fig 8A). This coincided with a slow and gradual change in the ion concentrations (Fig 8D–8G) and ionic reversal potentials (Fig 8B and 8C) due to predominantly passive leakage over the membrane.

At about $t$ = 48 s, the membrane potential reached the firing threshold, at which point the active channels started to use what was left of the concentration gradients to generate action potentials and $Ca^{2+}$ spikes. This resulted in a burst of activity. During this bursts of activity, the concentration gradients dissipated even faster, since both active and passive channels were then open. As a consequence, the "resting" membrane potential was further depolarized and the neuron went into depolarization block [65]. After this, the neuron continued to "leak" until it settled at a new steady state. The non-zero final equilibrium potential is known as the Donnan equilibrium or the Gibbs-Donnan equilibrium [69]. The reason why the cell did not approach an equilibrium with $\phi_m$ = 0 and identical ion concentrations on both side of the membrane, is that the model contained static residual charges, representing negatively charged macromolecules typically residing in the intracellular environment (see Methods), the sum of which resulted in a final state with a negatively charged inside. In addition, since the $Ca^{2+}$ channel inactivated, and since the model had no passive $Ca^{2+}$ leakage, $Ca^{2+}$ could end up being trapped inside/outside the membrane and did not by necessity approach the Donnan equilibrium, although it was close to it.

As the $Ca^{2+}$ dynamics in Fig 8G may seem counterintuitive, we here give some additional explanation of it. During the burst and initial stages of the depolarization block, the dendritic $Ca^{2+}$ channels were open. Extracellular $Ca^{2+}$ then diffused from the soma towards the dendrite, where it flowed into the neuron. This resulted in a low $Ca^{2+}$ concentration in both extracellular compartments and a high $Ca^{2+}$ concentration in the intracellular dendritic compartment. The reason why the intracellular $Ca^{2+}$ equilibrated more slowly than the extracellular, was that, by assumption, only 1% of the intracellular $Ca^{2+}$ concentration was unbuffered and free to diffuse (see Methods), hence, the effective intracellular concentration gradient was a factor 100 lower than it "appears" in Fig 8G.

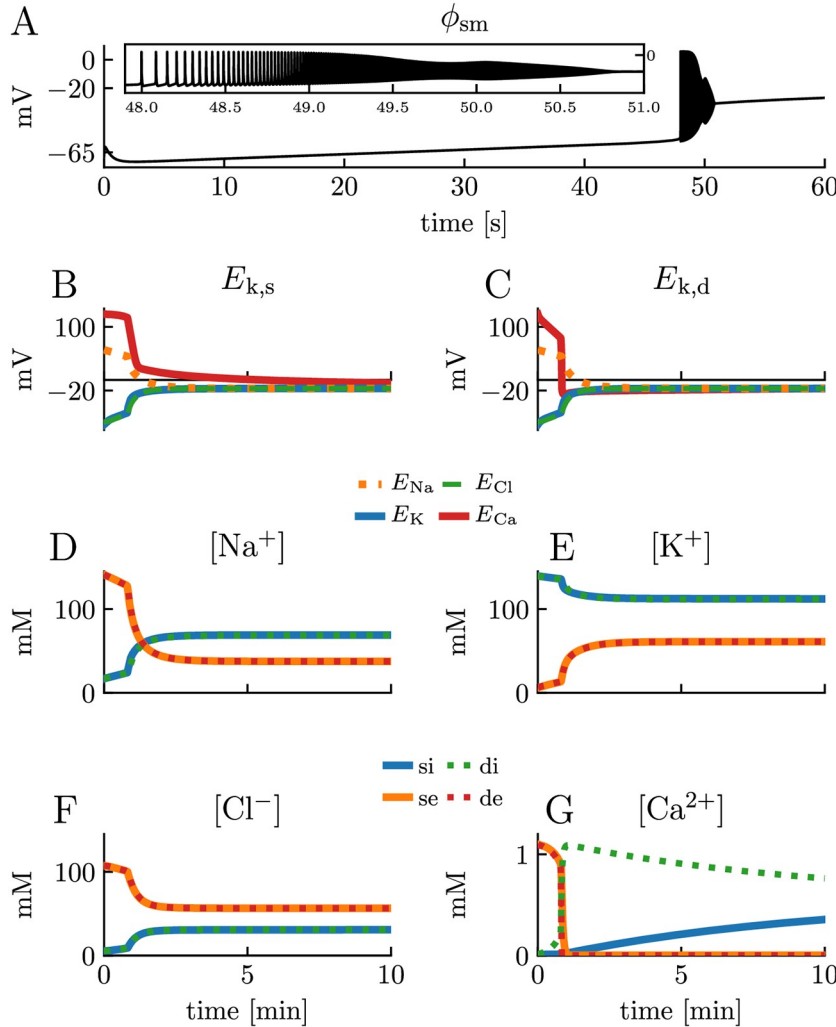

**Fig 8. The wave of death.** Simulation on the edPR model when the $3Na^+/2K^+$ pumps and the $Ca^{2+}/2Na^+$ exchangers were turned off. The model received no external stimulus. The simulation covered 10 minutes of biological time. **(A)** A 60 s sample of the dynamics of the somatic membrane potential $\phi_{sm}$. Inset shows a close-up of the burst of activity occurring at about $t$ = 48 s. **(B-C)** Reversal potentials in the soma **(B)** and dendrite **(C)**. **(D-G)** Ion concentrations in all four compartments. Somatic and dendritic concentrations were almost identical for all ion species except for $Ca^{2+}$. Indices $i$, $e$, s, and $d$ indicate *intracellular, extracellular, soma*, and *dendrite*, respectively. See the Parameterizations section in Methods for a full description of the parameters used.

A pattern resembling that in Fig 8A, i.e., a period of silence, followed by a burst of activity, and then silence again, has been seen in experimental EEG recordings of decapitated rats [70], where the activity burst was referred to as "the wave of death", and the phenomenon was ascribed to the lack of energy supply to homeostatic mechanisms. The simulation in Fig 8A represents the single-cell correspondence to this death wave. We note that this phenomenon has been simulated and analyzed thoroughly in a previous modeling study, using a simpler, single compartmental model with ion conservation [40]. We, therefore, do not analyze it further here.

## Loss in accuracy when neglecting electrodiffusive effects on concentration dynamics

The concentration-dependent effects studied in the previous subsection were predominantly due to changes in ionic reversal potentials. Effects like this could, therefore, be accounted for

by any model that in some way incorporates ion concentration dynamics [27–29, 33–57], provided that the ion concentration dynamics is accurately modeled. As we argued in the Introduction, previous multicompartmental neuron models that do incorporate ion concentration dynamics have not done it in a complete, ion conserving way that ensures a biophysically consistent relationship between ion concentration, electrical charge, and electrical potentials (see, e.g., [27, 48–57]). To specify, the change in the ion concentration in a given compartment will, in reality, depend on (i) the transmembrane influx of ions into this compartment, (ii) the diffusion of ions between this compartment and its neighboring compartment(s), and (iii) the electrical drift of ions between this compartment and its neighboring compartment(s). Some of the cited models account for only (i) [27, 49, 51], others account for (i) and (ii) [48, 50, 52–57], but neither account for (iii). When (iii) is not accounted for, electrical and diffusive processes are implicitly treated as independent processes, a simplifying assumption which is also incorporated in the reaction-diffusion module [71] in the NEURON simulation environment [72]. In models that apply this assumption, there will therefore be drift currents (along axons and dendrites) that affect $\phi_m$ (through the cable equation), but not the ion concentration dynamics, although they should, since also the drift currents are mediated by ions.

Here, we use simulations on the edPR model to test the inaccuracy introduced when not accounting for the effect of drift currents on ion concentration dynamics. We do so by comparing how many ions that were transferred from the somatic to the dendritic compartment through the intracellular (Fig 9A) and extracellular (Fig 9B) space, due to ionic diffusion (orange curves) versus electrical drift (blue curves), throughout the simulation in Fig 3. We note that Fig 9 shows the accumulatively moved number of ions (from time zero to $t$) due to axial fluxes exclusively. That is, the large number of, for example, Na$^+$ ions transported intracellularly from the dendrite to the soma (negative sign) in Fig 9A1, does not by necessity mean that Na$^+$ ions were piling up in the soma compartment, as the membrane efflux of Na$^+$ was not accounted for in the figure.

Although diffusion tended to dominate the intracellular transport of ions on the long time scale (Fig 9A1–9A4), the transport due to electrical drift was not vanishingly small. For example, the number of K$^+$ and Cl$^-$ ions transported by electrical drift was at the end of the stimulus period ($t$ = 20 s) about 35% of the transport due to diffusion for both species. In the extracellular space, diffusion was the clearly dominant transporter of Na$^+$ and K$^+$ (Fig 9B1 and 9B2), while diffusion and electrical drift were of comparable magnitude for the other ion species (Fig 9B3–9A4). Of course, these estimates are all specific to the edPR model, as they will depend strongly on the included ion channels, ion pumps and cotransporters, and on how they are distributed between the soma and dendrite. In general, however, the simulations in Fig 9 suggest that electrical drift is likely to have a non-negligible effect on ion concentration dynamics, and that ignoring this effect will give rise to rather inaccurate estimates.

Finally, we also converted the sum of ionic fluxes in Fig 9 into an effective current, represented as the number of transported unit charges, e$^+$ (Fig 9A5–9B5). Interestingly, diffusion and drift contributed almost equally to the axial charge transport in the system. We note, however, that the movement of charges per time unit is indicated by the slope of the curves, which was much larger for the drift case (blue curve) than for diffusion (orange curve). The drift curve had a jigsaw shape, which shows that drift was moving charges back and forth in the system, while the diffusion always went in the same direction, explaining why it, despite being smaller than the drift current, had a comparably large accumulative effect on charge transport. The temporally averaged picture of charge transport that emerges from Fig 9A5 is that of a slow current loop where charge is transferred intracellularly from the soma to the dendrite (Fig 9A5), where it crosses the membrane (outward current), and then is transferred extracellularly back from the dendrite to the soma (Fig 9B5), before crossing the membrane again

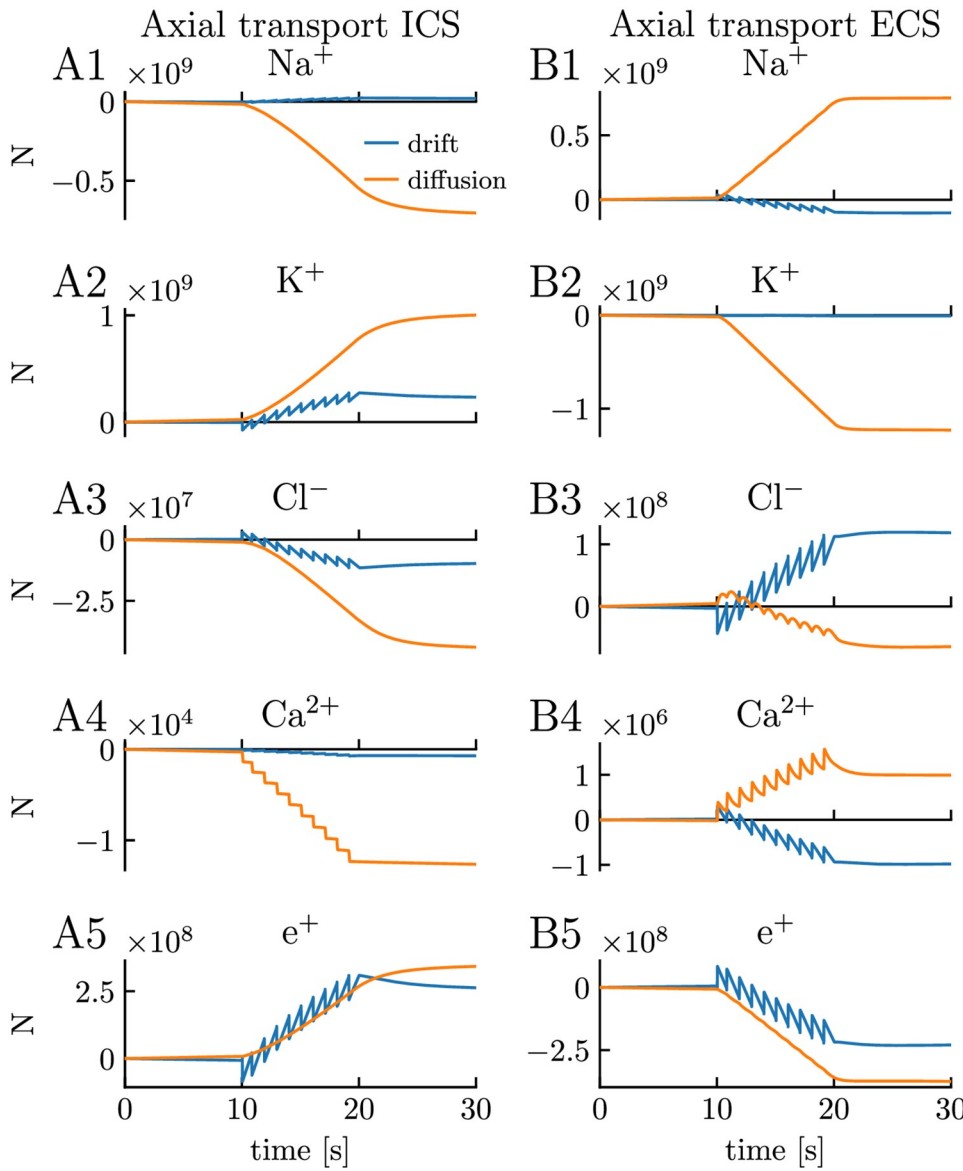

**Fig 9. Axial transport of ions and charge due to drift versus diffusion. (A1-A4)** The number of ions transported intracellularly from soma to dendrite from time zero to *t* by electrical drift versus ionic diffusion. **(B1-B4)** The number of ions transported extracellularly from (outside) soma to (outside) dendrite from time zero to *t*. **(A5)** Net charge transported intracellularly from soma to dendrite, represented as the number of unit charges e⁺. **(B5)** Net charge transported extracellularly from soma to dendrite, represented as the number of unit charges e⁺. **(A-B)** The simulation was the same as in Fig 3. See the Analysis section in Methods for a description of how we did the calculations.

(inward current). This configuration is similar to the slow loop current seen during spatial buffering by astrocytes (see, e.g. Fig 1 in [67]).

## Loss in accuracy when neglecting electrodiffusive effects on voltage dynamics

In the previous section, we investigated the consequences of neglecting (iii) the contribution of drift currents on ion concentration dynamics. Here, we investigate the consequences of neglecting the effect of ionic diffusion (along dendrites) on the electrical potential, focusing on

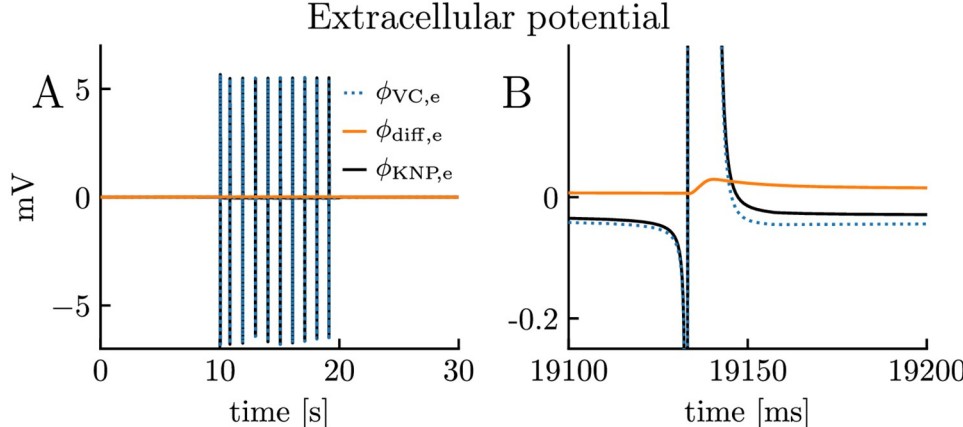

**Fig 10. Effect of diffusion on extracellular potential.** The extracellular potential $\phi_e$ in the edPR model, split (cf. Eq 81) into a component explained by standard VC-theory ($\phi_{VC,e}$) and a "correction" ($\phi_{diff,e}$) when diffusive contributions are accounted for. **(A-B)** The simulation was the same as in Fig 3. **(B)** Close-up of selected AP in **(A)**. See the Analysis section in Methods for a description of how we calculated $\phi_{VC,e}$ and $\phi_{diff,e}$.

the extracellular potential $\phi_e$. Forward modeling of extracellular potentials is typically based on volume conductor (VC) theory [16–18, 20, 21], which assumes that diffusive effects on electrical potentials are negligible. Being based on a unified electrodiffusive KNP framework (Fig 1), the edPR model accounts for the effects of ionic diffusion on the electrical potentials, and can thus be used to address the validity of this assumption.

To illustrate the effect of diffusion on $\phi_e$, we may split it into a component $\phi_{VC,e}$ explained by standard VC-theory, and a component $\phi_{diff,e}$ representing the additional contribution caused by diffusive currents (Eq 81). In the simulation in Fig 3, the diffusive contribution was found to be very small compared to the VC-component (Fig 10). However, while $\phi_{VC,e}$ fluctuated rapidly from negative to positive values during neuronal activity, $\phi_{diff,e}$ varied on a slower time scale and had the same directionality throughout the simulation. This is equivalent to what we saw in Fig 9B5, i.e., that diffusion always moved charge in the same direction. Moreover, if we take the temporal averages of the potentials over the time series in Fig 10A, we find that they are -0.0023 mV, 0.0037 mV, and -0.0060 mV for $\phi_e$, $\phi_{diff,e}$, and $\phi_{VC,e}$, respectively. This shows that the average diffusion- and VC-components of the total potential were of the same order of magnitude. As we also have demonstrated in previous studies, diffusion is thus likely to be important for the low-frequency components of extracellular potentials [31, 32, 73, 74]. Albeit small, the slowly varying diffusion evoked shifts in $\phi_e$ are putatively important for explaining the direct-current (DC) like shifts and long-time concentration dynamics reported during, e.g., spreading depression [25, 26].

## Discussion

The original Pinsky-Rinzel (PR) is a reduced model of a hippocampal neuron, which reproduces the essential somatodendritic firing properties of CA3 neurons despite having only two compartments [3]. Simplified neuron models like that are useful, partly because their reduced complexity makes them easier to analyze, and as such, can lead to insight in key neuronal mechanisms, and partly because they demand less computer power and can be used as modules in large scale network simulations. Whereas the PR model, as most available neuron models, assumes that ion concentrations remain constant during the simulated period, the electrodiffusive Pinsky-Rinzel (edPR) model proposed here models ion concentration

dynamics explicitly. The edPR model may thus be seen as a supplement to the PR model, which should be applied to simulate conditions where ion concentrations are expected to vary with time.

In the results section, we showed that the edPR model closely reproduced the firing properties of the PR model for short term dynamics (Fig 5), and for long term dynamics provided that the firing rate was sufficiently low for the homeostatic mechanisms to maintain ion concentrations close to baseline (Fig 6). We also showed that if the firing rate became too high (Fig 7), or if the homeostatic mechanisms were impaired (Fig 8), unsuccessful homeostasis would cause ion concentrations to gradually shift over time, and lead to slowly developing changes in the firing properties of the edPR model, changes that were not accounted for by the original PR model. The edPR model was based on an electrodiffusive framework [60], which ensured a consistent relationship between ion concentrations, electrical charge, and electrical potential in four compartments. To our knowledge, the edPR model is the first multicompartmental neuronal model that ensures complete and consistent ion concentration and charge conservation.

## Model assumptions

The construction of the edPR model naturally involved making a set of modeling choices, and the most important of these are discussed here. Firstly, in the construction of the model, we focused on morphological simplicity, biophysical rigor, and mechanistic understanding, rather than on replicating any specific biological scenario and incorporating biological details. Secondly, simultaneous data of variations in all intra- and extracellular concentrations during neuronal firing are not available, and it might not even be feasible to obtain such data. Consequently, computational modeling based on biophysical constraints may be the best means to estimate it. The concentration dynamics in the edPR model were thus not directly constrained to data but constrained so that there was, at all times, an internally consistent relationship between all ion concentrations and all electrical potentials, ensuring an electroneutral bulk solution. Thirdly, to include extracellular dynamics to models of neurons or networks of such is computationally challenging, since the extracellular space, in reality, is an un-confined three-dimensional continuum, locally affected by populations of nearby neurons and glial cells. As we wanted to keep things simple and conceptual, we chose to use closed boundary conditions, i.e., no ions and no charge were allowed to leave or enter the system consisting of the single (2-compartment) neuron and its local and confined (2-compartment) surrounding (Fig 2). Tecnically, it would be straightforward to increase the number of compartments (i.e., the spatial resolution) in the model.

A consequence of using closed boundary conditions was that the extracellular (like the intracellular) currents became one-dimensional (from soma to dendrite), while in reality, extracellular currents pass through a three-dimensional volume conductor. The edPR model could be made three dimensional if embedded in a bi- or tri-domain model (as discussed below). However, currently, it is 1D, and the effect of the 1D assumption was essentially an increase in the total resistance (fewer degrees of freedom) for extracellular currents, which gave rise to an artificially high amplitude in extracellular AP signatures (Fig 3). We note, however, that the closed boundary is actually equivalent to assuming periodic boundary conditions, so that the edPR model essentially simulates the hypothetical case of a population of perfectly synchronized neurons, i.e., one where all neurons are doing exactly the same as the simulated neuron, so that no spatial variation occurs. Likely, this may give accurate predictions for ion concentration shifts over time, as these reflect a temporal average of activity, but less accurate predictions for brief and unique electrical events, such as action potentials, which are not likely to be elicited in perfect synchrony by a large population [31].

Fourthly, to faithfully represent a morphologically complex neuron with a reduced number of compartments is a non-trivial task. Available analytical theory for collapsing branching dendrites into equivalent cylinders are generally based on certain assumptions about branching symmetries, and on preserving electrotonic distances [75]. However, it is unlikely that the length constants of electrodynamics and ion concentration dynamics scale in the same way. Hence, in the edPR model, the volumes and membrane areas of, and cross-section areas between, the two neuronal compartments were here introduced as rather arbitrary model choices, fixed at values that were verified to give agreement between the firing properties of the edPR model and the PR model.

## Outlook

Being applicable to simulate conditions with failed homeostasis, the edPR model opens up for simulating a range of pathological conditions, such as spreading depression or epilepsy [22–25], which are associated with large scale shifts in extracellular ion concentrations. A particular context in which we anticipate the edPR model to be useful is that of simulating spreading depression. Previous spatial, electrodiffusive, and biophysically consistent models of spreading depression have targeted the problem at a large-scale tissue-level, using a mean-field approach [30, 76, 77]. These models were inspired by the *bi-domain* model [78], which has been successfully applied in simulations of cardiac tissue [79, 80]. The bi-domain model is a coarse-grained model, in which the tissue is considered as a bi-phasic continuum consisting of an intracellular and extracellular domain. That is, a set of intra- and extracellular variables (i.e., voltages and ion concentrations), and the ionic exchange between the intra- and extracellular domains, are defined at each point in space. This simplification allows for large scale simulations of signals that propagate through tissue but sacrifices morphological detail. In the context of spreading depression, a shortcoming with this simplification is that the leading edge of the spreading depression wave in both the hippocampus and cortex is in the layers containing the apical dendrites [22]. This suggests that the different expression of membrane mechanisms in deeper (somatic) and higher (dendritic) layers may be crucial for fully understanding the propagation and genesis of the wave. In this context, the edPR model could enter as a module in a, let us say, *bi-times-two-domain* model, where each point in ($xy$) space contains a set of (i) somatic intracellular variables, (ii) somatic extracellular variables, (iii) dendritic intracellular variables, and (iv) dendritic extracellular variables, and thus accounts for the differences between the higher and lower layers. We should note that the state of the art models of spreading depression are not bi-domain models but rather tri-domain models, as they also include a glial domain to account especially for the work done by astrocytes in K$^+$ buffering [30, 76, 77]. Hence, to use the edPR model to expand the current spreading depression models, a natural first step would be to include a glial (astrocytic) compartment in it, so that it eventually could be implemented as a *tri-times-two-domain* model.

## Methods

### The Kirchoff-Nernst-Planck (KNP) framework

In the following section, we derive the KNP continuity equations for a one-dimensional system containing two plus two compartments (Fig 2A), with sealed boundary conditions (i.e., no ions can enter or leave the system). The geometrical parameters used in the edPR model were as defined in Table 1. Since typical neuronal/extracellular/glial volume fractions in neuronal tissue are 0.4/0.2/0.4 [82], we let the extracellular space be half as voluminous as the intracellular neuronal space.

**Table 1. Geometrical parameters.**

| Parameter | Value |
|---|---|
| $\Delta x$ (distance between the two compartments) | $667 \cdot 10^{-6}$ m |
| $A_s$ (somatic membrane area) | $616 \cdot 10^{-12}$ m$^2$ * |
| $A_d$ (dendritic membrane area) | $616 \cdot 10^{-12}$ m$^2$ * |
| $A_i$ (intracellular cross-section area) | $\alpha \cdot A_s$ † |
| $A_e$ (extracellular cross-section area) | $A_i/2$ |
| $V_{si}$ (somatic intracellular volume) | $1437 \cdot 10^{-18}$ m$^3$ * |
| $V_{di}$ (dendritic intracellular volume) | $1437 \cdot 10^{-18}$ m$^3$ * |
| $V_{se}$ (somatic extracellular volume) | $718.5 \cdot 10^{-18}$ m$^3$ * |
| $V_{de}$ (dendritic extracellular volume) | $718.5 \cdot 10^{-18}$ m$^3$ * |

* The intracellular volumes ($V_{si}$, $V_{di}$) and membrane areas ($A_s$, $A_d$) correspond to spheres with radius 7 µm.
† The parameter $\alpha$ describes the coupling strength of the model and is defined in the Parameterizations section. Its default value was 2.

Two kinds of fluxes are involved: transmembrane fluxes and intra- and extracellular fluxes. The framework is general to the choice of the transmembrane fluxes. A transmembrane flux of ion species k ($j_{k,m}$) represents the sum of all fluxes through all membrane mechanisms that allow ion k to cross the membrane.

Intracellular flux densities are described by the Nernst-Planck equation:

$$j_{k,i} = -\frac{D_k}{\lambda_i^2} \frac{\gamma_k([k]_{di} - [k]_{si})}{\Delta x} - \frac{D_k z_k F}{\lambda_i^2 RT} \overline{[k]}_i \frac{\phi_{di} - \phi_{si}}{\Delta x}. \tag{1}$$

In Eq 1, $D_k$ is the diffusion constant, $\gamma_k$ is the fraction of freely moving ions, that is, ions that are not buffered or taken up by the ER, $\lambda_i$ is the tortuosity, which represents the slowing down of diffusion due to obstacles, $\gamma_k([k]_{di}-[k]_{si})/\Delta x$ is the axial concentration gradient, $z_k$ is the charge number of ion species k, $F$ is the Faraday constant, $R$ is the gas constant, $T$ is the absolute temperature, $\overline{[k]}_i$ is the average concentration, that is, $\gamma_k([k]_{di} + [k]_{si})/2$, and $(\phi_{di}-\phi_{si})/\Delta x$ is the axial potential gradient. Similarly, the extracellular flux densities are described by

$$j_{k,e} = -\frac{D_k}{\lambda_e^2} \frac{[k]_{de} - [k]_{se}}{\Delta x} - \frac{D_k z_k F}{\lambda_e^2 RT} \overline{[k]}_e \frac{\phi_{de} - \phi_{se}}{\Delta x}. \tag{2}$$

In Eq 2, we do not include $\gamma_k$, as all ions can move freely in the extracellular space. Diffusion constants, tortuosities, and intracellular fractions of freely moving ions used in the edPR model were as in Table 2.

**Ion conservation.** The KNP framework is based on the constraint of ion conservation. To keep track of ion concentrations we solve four differential equations for each ion species k:

$$\frac{d[k]_{si}}{dt} = -j_{k,sm} \cdot \frac{A_s}{V_{si}} - j_{k,i} \cdot \frac{A_i}{V_{si}}, \tag{3}$$

$$\frac{d[k]_{di}}{dt} = -j_{k,dm} \cdot \frac{A_d}{V_{di}} + j_{k,i} \cdot \frac{A_i}{V_{di}}, \tag{4}$$

$$\frac{d[k]_{se}}{dt} = +j_{k,sm} \cdot \frac{A_s}{V_{se}} - j_{k,e} \cdot \frac{A_e}{V_{se}}, \tag{5}$$

**Table 2. Diffusion constants, tortuosities, and intracellular fractions of freely moving ions.**

| Parameter | Value | Reference |
|---|---|---|
| $D_{Na}$ (Na$^+$ diffusion constant) | $1.33 \cdot 10^{-9}$ m$^2$/s | [31, 81] |
| $D_k$ (K$^+$ diffusion constant) | $1.96 \cdot 10^{-9}$ m$^2$/s | [31, 81] |
| $D_{Cl}$ (Cl$^-$ diffusion constant) | $2.03 \cdot 10^{-9}$ m$^2$/s | [31, 81] |
| $D_{Ca}$ (Ca$^{2+}$ diffusion constant) | $0.71 \cdot 10^{-9}$ m$^2$/s | [31, 81] |
| $\lambda_i$ (intracellular tortuosity) | 3.2 | [60, 82] |
| $\lambda_e$ (extracellular tortuosity) | 1.6 | [60, 82] |
| $\gamma_{Na}, \gamma_K, \gamma_{Cl}$ (intracellular fractions of free ions) | 1 | |
| $\gamma_{Ca}$ (intracellular fraction of free ions) | 0.01 | |

$$\frac{d[k]_{de}}{dt} = +j_{k,dm} \cdot \frac{A_d}{V_{de}} + j_{k,e} \cdot \frac{A_e}{V_{de}}. \tag{6}$$

For each compartment, all flux densities are multiplied by the area they go through and divided by the volume they enter to calculate the change in ion concentration. If we insert the Nernst-Planck equation (Eq 1) for the intracellular flux density, the first of these equations becomes:

$$\frac{d[k]_{si}}{dt} = -j_{k,sm} \cdot \frac{A_s}{V_{si}} + \frac{A_i D_k}{V_{si}\lambda_i^2 \Delta x}\left[\gamma_k([k]_{di} - [k]_{si}) + \frac{z_k F}{RT}\overline{[k]}_i(\phi_{di} - \phi_{si})\right], \tag{7}$$

where the voltage variables so far are undefined.

**Four constraints to derive $\phi$.** If we have four ion species (Na$^+$, K$^+$, Cl$^-$, and Ca$^{2+}$) in four compartments, we have 20 unknown parameters (16 for [k] and four for $\phi$), while Eqs 3–6 for four ion species give us only 16 equations. To solve this, we need to define additional constraints that allow us to express the potentials $\phi$ in terms of ion concentrations.

As we may define an arbitrary reference point for $\phi$, we take

$$\phi_{de} = 0, \tag{8}$$

as our first constraint, i.e., (i) the potential outside the dendrite is defined to be zero.

The second constraint is that (ii) the membrane is a parallel plate capacitor that always separates a charge $Q$ on one side from an opposite charge $-Q$ on the other side, giving rise to a voltage difference

$$\phi_m = Q/C_m. \tag{9}$$

Here, $C_m$ is the total capacitance of the membrane, i.e., $C_m = c_m A_m$, where $c_m$ is the more commonly used capacitance per membrane area. As, by definition, $\phi_m = \phi_i - \phi_e$, we get:

$$\phi_{dm} = \phi_{di} = Q_{di}/C_m, \tag{10}$$

in the dendrite (since $\phi_{de} = 0$), and

$$\phi_{sm} = \phi_{si} - \phi_{se} = Q_{si}/C_m, \tag{11}$$

in the soma.

The third constraint is that (iii) we assume bulk electroneutrality. This means that the net charge associated with the ion concentrations in a given compartment by constraint must be identical to the membrane charge in this compartment. The intracellular dendritic charge is

thus $Q_{di} = F\sum_k z_k[k]_{di}V_{di}$. By inserting this into Eq 10, we obtain the final expression for $\phi_{di}$:

$$\phi_{di} = (F\sum_k z_k[k]_{di}V_{di})/(c_m A_d). \tag{12}$$

By inserting the equivalent expression for $Q_{si}$ into Eq 11, we get

$$\phi_{si} - \phi_{se} = Q_{si}/C_m = (F\sum_k z_k[k]_{si}V_{si})/(c_m A_s). \tag{13}$$

Here, the extracellular potential is not set to zero, so we need a fourth constraint to determine $\phi_{si}$ and $\phi_{se}$ separately.

The fourth and final constraint is that (iv) we must ensure charge anti-symmetry. For the charge anti-symmetry between the two sides of the capacitive membrane ($Q_i = -Q_e$) to be preserved in time, we must define our initial conditions so that this is the case at $t = 0$, and the system dynamics so that this stays the case. Hence, the system dynamics must ensure that $dQ_{di}/dt = -dQ_{de}/dt$ and $dQ_{si}/dt = -dQ_{se}/dt$. The membrane currents (in isolation) will always fulfill this criterion, as any charge that crosses the membrane by definition disappears from one side of it and pops up at the other. Hence, we thus need to make sure that also the axial currents (in isolation) fulfill the criterion. The system must thus be constrained so that, if an extracellular current transports a charge $\delta q$ into a given extracellular compartment, the intracellular current must transport the opposite charge $-\delta q$ into the adjoint intracellular compartment. That is, we must have that:

$$A_i i_i = -A_e i_e, \tag{14}$$

where $i_i$ and $i_e$ are the intra- and extracellular current densities, respectively. To find an expression for these, we multiply Eqs 1 and 2 by $Fz_k$ and sum over all ion species k. The expressions for the intra- and extracellular current densities then become:

$$i_i = -\frac{F}{\lambda_i^2 \Delta x}\sum_k D_k z_k \gamma_k([k]_{di} - [k]_{si}) - \frac{F^2}{RT\lambda_i^2 \Delta x}\sum_k D_k z_k^2 \overline{[k]}_i(\phi_{di} - \phi_{si}), \tag{15}$$

$$i_e = -\frac{F}{\lambda_e^2 \Delta x}\sum_k D_k z_k([k]_{de} - [k]_{se}) - \frac{F^2}{RT\lambda_e^2 \Delta x}\sum_k D_k z_k^2 \overline{[k]}_e(\phi_{de} - \phi_{se}). \tag{16}$$

In Eq 15, the first term is the diffusion current density:

$$i_{diff,i} = -\frac{F}{\lambda_i^2 \Delta x}\sum_k D_k z_k \gamma_k([k]_{di} - [k]_{si}), \tag{17}$$

which is defined by the ion concentrations. The second term is the field driven current density

$$i_{field,i} = -\sigma_i \frac{(\phi_{di} - \phi_{si})}{\Delta x}, \tag{18}$$

where we have identified the conductivity as

$$\sigma_i = \frac{F^2}{RT\lambda_i^2}\sum_k D_k z_k^2 \overline{[k]}_i. \tag{19}$$

Similarly, Eq 16 can be written in terms of $i_{\text{diff,e}}$, $i_{\text{field,e}}$, and $\sigma_{\text{e}}$. By combining Eqs 14, 15 and 16, we obtain:

$$-A_{\text{i}}i_{\text{diff,i}} + A_{\text{i}}\sigma_{\text{i}} \cdot \frac{(\phi_{\text{di}} - \phi_{\text{si}})}{\Delta x} = A_{\text{e}}i_{\text{diff,e}} - A_{\text{e}}\sigma_{\text{e}} \cdot \frac{(\phi_{\text{de}} - \phi_{\text{se}})}{\Delta x}. \tag{20}$$

In Eq 20, $\phi_{\text{di}}$ and $\phi_{\text{de}}$ are already known from Eqs 8 and 12, while $i_{\text{diff}}$ and $\sigma$ are expressed in terms of ion concentrations. We may thus solve Eqs 13 and 20 for the last two voltage variables $\phi_{\text{se}}$ and $\phi_{\text{si}}$:

$$\phi_{\text{se}} = \left( \phi_{\text{di}} - \frac{\Delta x}{\sigma_{\text{i}}} \cdot i_{\text{diff,i}} - \frac{A_{\text{e}}\Delta x}{A_{\text{i}}\sigma_{\text{i}}} \cdot i_{\text{diff,e}} - \frac{Q_{\text{si}}}{c_{\text{m}}A_{\text{s}}} \right) \Big/ \left( 1 + \frac{A_{\text{e}}\sigma_{\text{e}}}{A_{\text{i}}\sigma_{\text{i}}} \right), \tag{21}$$

$$\phi_{\text{si}} = \frac{Q_{\text{si}}}{c_{\text{m}}A_{\text{s}}} + \phi_{\text{se}}. \tag{22}$$

## Membrane mechanics

**Leakage channels.** In the original PR model, the membrane leak current represents the combined contribution from all ion species. When using the KNP framework, on the other hand, where we keep track of all ions separately, the leak current must be ion-specific. We modeled this as in [45], that is, for each ion species k, we implemented a passive flux density across the membrane

$$j_{\text{k,leak}} = \overline{g}_{\text{k,leak}}(\phi_{\text{m}} - E_{\text{k}})/(Fz_{\text{k}}), \tag{23}$$

where $\overline{g}_{\text{k,leak}}$ is the ion conductance, $\phi_{\text{m}}$ is the membrane potential, $E_{\text{k}}$ is the reversal potential, $F$ is the Faraday constant, and $z_{\text{k}}$ is the charge number. The reversal potential is a function of ion concentrations, and is calculated using the Nernst equation:

$$E_{\text{k}} = \frac{RT}{z_{\text{k}}F} \ln \frac{[\text{k}]_{\text{e}}}{\gamma_{\text{k}}[\text{k}]_{\text{i}}}. \tag{24}$$

Here, $R$ is the gas constant, $T$ is the absolute temperature, $\gamma_{\text{k}}$ is the intracellular fraction of free ions, and $[\text{k}]_{\text{e}}$ and $[\text{k}]_{\text{i}}$ are the concentrations of ion k outside and inside the cell, respectively. We included $\text{Na}^+$, $\text{K}^+$, and $\text{Cl}^-$ leak currents in both compartments.

**Active ion channels.** All active ion channel currents were adopted from the original PR model [3], as they were described in [8], and converted to ion channel fluxes. The soma compartment contained a $\text{Na}^+$ flux ($j_{\text{Na}}$) and a $\text{K}^+$ delayed rectifier flux ($j_{\text{K-DR}}$), while the dendrite contained a voltage-dependent $\text{Ca}^{2+}$ flux ($j_{\text{Ca}}$), a voltage-dependent $\text{K}^+$ AHP flux ($j_{\text{K-AHP}}$), and a $\text{Ca}^{2+}$-dependent $\text{K}^+$ flux ($j_{\text{K-C}}$):

$$j_{\text{Na}} = g_{\text{Na}}(\phi_{\text{sm}} - E_{\text{Na,s}})/(Fz_{\text{Na}}), \tag{25}$$

$$j_{\text{K-DR}} = g_{\text{DR}}(\phi_{\text{sm}} - E_{\text{K,s}})/(Fz_{\text{K}}), \tag{26}$$

$$j_{\text{Ca}} = g_{\text{Ca}}(\phi_{\text{dm}} - E_{\text{Ca,d}})/(Fz_{\text{Ca}}), \tag{27}$$

$$j_{\text{K-AHP}} = g_{\text{AHP}}(\phi_{\text{dm}} - E_{\text{K,d}})/(Fz_{\text{K}}), \tag{28}$$

$$j_{K-C} = g_C(\phi_{dm} - E_{K,d})/(Fz_K).\tag{29}$$

The voltage-dependent conductances were modeled using the Hodkin-Huxley formalism with differential equations for the gating variables:

$$\frac{dx}{dt} = \alpha_x(1-x) - \beta_x x, \quad \text{with } x = m, h, n, s, c, q,\tag{30}$$

$$\frac{dz}{dt} = \frac{z_\infty - z}{\tau_z}.\tag{31}$$

The conductances and gating variables were given by:

$$g_{Na} = \overline{g}_{Na} m_\infty^2 h,\tag{32}$$

$$g_{DR} = \overline{g}_{DR} n,\tag{33}$$

$$g_{Ca} = \overline{g}_{Ca} s^2 z,\tag{34}$$

$$g_C = \overline{g}_C c\chi([Ca^{2+}]),\tag{35}$$

$$g_{AHP} = \overline{g}_{AHP} q,\tag{36}$$

$$\alpha_m = -\frac{3.2 \cdot 10^5 \cdot \phi_1}{\exp(-\phi_1/0.004) - 1}, \text{ with } \phi_1 = \phi_m + 0.0469\tag{37}$$

$$\beta_m = \frac{2.8 \cdot 10^5 \cdot \phi_2}{\exp(\phi_2/0.005) - 1}, \text{ with } \phi_2 = \phi_m + 0.0199\tag{38}$$

$$m_\infty = \frac{\alpha_m}{\alpha_m + \beta_m}\tag{39}$$

$$\alpha_h = 128 \exp\frac{-0.043 - \phi_m}{0.018},\tag{40}$$

$$\beta_h = \frac{4000}{1 + \exp(-\phi_3/0.005)}, \text{ with } \phi_3 = \phi_m + 0.02\tag{41}$$

$$\alpha_n = -\frac{1.6 \cdot 10^4 \cdot \phi_4}{\exp(-\phi_4/0.005) - 1}, \text{ with } \phi_4 = \phi_m + 0.0249\tag{42}$$

$$\beta_n = 250 \exp(-\phi_5/0.04), \text{ with } \phi_5 = \phi_m + 0.04\tag{43}$$

$$\alpha_s = \frac{1600}{1 + \exp(-72(\phi_m - 0.005))},\tag{44}$$

$$\beta_s = \frac{2 \cdot 10^4 \cdot \phi_6}{\exp(\phi_6/0.005) - 1}, \text{ with } \phi_6 = \phi_m + 0.0089\tag{45}$$

$$z_\infty = \frac{1}{1 + \exp\left(\phi_7/0.001\right)}, \quad \text{with } \phi_7 = \phi_m + 0.03 \tag{46}$$

$$\tau_z = 1, \tag{47}$$

$$\alpha_c = \begin{cases} 52.7\exp\left(\dfrac{\phi_8}{0.011} - \dfrac{\phi_9}{0.027}\right), & \text{if } \phi_m \leq -0.01 \text{ V} \\[2mm] 2000\exp(-\phi_9/0.027), & \text{otherwise} \end{cases} \tag{48}$$

$$\text{with } \phi_8 = \phi_m + 0.05 \text{ and } \phi_9 = \phi_m + 0.0535 \tag{49}$$

$$\beta_c = \begin{cases} 2000\exp\left(-\phi_9/0.027\right) - \alpha_c, & \text{if } \phi_m \leq -0.01 \text{ V} \\[2mm] 0, & \text{otherwise} \end{cases} \tag{50}$$

$$\chi = \min\left(\frac{\gamma_{Ca}[Ca^{2+}] - 99.8 \cdot 10^{-6}}{2.5 \cdot 10^{-4}}, 1\right), \tag{51}$$

$$\alpha_q = \min\left(2 \cdot 10^4(\gamma_{Ca}[Ca^{2+}] - 99.8 \cdot 10^{-6}), 10\right), \tag{52}$$

$$\beta_q = 1. \tag{53}$$

All these equations were taken from [8] (with errata [83]) and converted so that values are given in SI units: units for rates ($\alpha$'s, $\beta$'s) are 1/s, unit for $\tau_z$ is s, and units for voltages $\phi$ are V. The equations were used in their original form, except those related to $Ca^{2+}$ dynamics, where we made the following changes: Firstly, as a large fraction of intracellular $Ca^{2+}$ is buffered or taken up by the ER, we multiplied $[Ca^{2+}]$ in Eqs 51 and 52 by a factor $\gamma_{Ca}$, which refers to the fraction of free $Ca^{2+}$ within the cell, and set this to be 0.01. As $[Ca^{2+}]$ in Eqs 51 and 52 were multiplied with 0.01, only the free $Ca^{2+}$ could affect the $Ca^{2+}$ activated ion channels. We further assumed that only the free $Ca^{2+}$ could move between the intracellular compartments (Eq 1) and affect the $Ca^{2+}$ reversal potential (Eq 24). Secondly, the original PR model had an abstract and unitless variable for the intracellular $Ca^{2+}$ concentration, with a basal concentration of 0.2, while we defined a (biophysically realistic) baseline concentration of 0.01 mM, which corresponds to a concentration of *free* $Ca^{2+}$ of 100 nM. In Eqs 51 and 52 we therefore subtracted $99.8 \cdot 10^{-6}$ (mol/m³) from the $Ca^{2+}$ concentration to correct for the shift in baseline. Thirdly, we modified the voltage-dependent $Ca^{2+}$ current to include an inactivation variable $z$ (Eqs 31 and 34). We implemented this inactivation like they did in [84] (Eqs A2-A3), but set the time constant to 1 s, the half-activation voltage to -30 mV, and the slope of the steady-state Boltzmann fit to $z_\infty$ to 0.001. In the original PR model, inactivation was neglected due to the argument that it was too slow to have an impact on simulation outcomes [2]. However, in our simulations, we observed that it had a significant impact, and therefore we included it.

**Homeostatic mechanisms.** To maintain baseline ion concentrations for low frequency activity we added $3Na^+/2K^+$ pumps, $K^+/Cl^-$ cotransporters (KCC2), and $Na^+/K^+/2Cl^-$

cotransporters (NKCC1). Their functional forms were taken from [45].

$$j_{\text{pump}} = \frac{\rho}{1.0 + \exp\left((25 - [\text{Na}^+]_i)/3\right)} \cdot \frac{1.0}{1.0 + \exp\left(3.5 - [\text{K}^+]_e\right)}, \tag{54}$$

$$j_{\text{kcc2}} = U_{\text{kcc2}} \ln\left(\frac{[\text{K}^+]_i[\text{Cl}^-]_i}{[\text{K}^+]_e[\text{Cl}^-]_e}\right), \tag{55}$$

$$j_{\text{nkcc1}} = U_{\text{nkcc1}} f([\text{K}^+]_e)\left(\ln\left(\frac{[\text{K}^+]_i[\text{Cl}^-]_i}{[\text{K}^+]_e[\text{Cl}^-]_e}\right) + \ln\left(\frac{[\text{Na}^+]_i[\text{Cl}^-]_i}{[\text{Na}^+]_e[\text{Cl}^-]_e}\right)\right), \tag{56}$$

$$f([\text{K}^+]_e) = \frac{1}{1 + \exp\left(16 - [\text{K}^+]_e\right)}, \tag{57}$$

where $\rho$, $U_{\text{kcc2}}$, and $U_{\text{nkcc1}}$ are pump and cotransporter strengths. We assumed optimal pump functionality and set $\rho$ to be the pump strength used in [45] for the fully oxygenated state with normal bath potassium ($\rho_{\text{max}}$).

Intracellular $\text{Ca}^{2+}$ decay was modeled in a similar fashion as in [3], but to ensure ion conservation we modeled it as an electroneutral $\text{Ca}^{2+}/2\text{Na}^+$ exchanger, exchanging one $\text{Ca}^{2+}$ (outward) for two $\text{Na}^+$ (inward). Putatively, this phenomenological model for the decay could represent the joint effect of several mechanisms in a real system, such as the $\text{Ca}^{2+}/3\text{Na}^+$ exchanger, a $\text{Ca}^{2+}$ leakage current, SERCA pumps, etc. The decay flux density was defined as:

$$j_{\text{Ca-dec}} = U_{\text{Ca-dec}}([\text{Ca}^{2+}]_i - [\text{Ca}^{2+}]_{i,b}) \cdot \frac{V_i}{A_m} \tag{58}$$

where $U_{\text{Ca-dec}}$ is the decay rate, and $[\text{Ca}^{2+}]_{i,b}$ is the basal $\text{Ca}^{2+}$ concentration, set to 0.01 mM.

**Model summary.** We summarize the model here for easy reference. In short, we solved four differential equations for all ion species k:

$$\frac{d[\text{k}]_{si}}{dt} = -j_{\text{k,sm}} \cdot \frac{A_s}{V_{si}} - j_{\text{k,i}} \cdot \frac{A_i}{V_{si}}, \tag{59}$$

$$\frac{d[\text{k}]_{di}}{dt} = -j_{\text{k,dm}} \cdot \frac{A_d}{V_{di}} + j_{\text{k,i}} \cdot \frac{A_i}{V_{di}}, \tag{60}$$

$$\frac{d[\text{k}]_{se}}{dt} = +j_{\text{k,sm}} \cdot \frac{A_s}{V_{se}} - j_{\text{k,e}} \cdot \frac{A_e}{V_{se}}, \tag{61}$$

$$\frac{d[\text{k}]_{de}}{dt} = +j_{\text{k,dm}} \cdot \frac{A_d}{V_{se}} + j_{\text{k,e}} \cdot \frac{A_e}{V_{de}}. \tag{62}$$

**Table 3. Temperature and physical constants.**

| Parameter | Value | Reference |
|---|---|---|
| $T$ (absolute temperature) | 309.14 K | [45]* |
| $F$ (Faraday constant) | $9.648 \cdot 10^4$ C/mol | |
| $R$ (gas constant) | 8.314 J/(mol.K) | |

* The temperature is not explicitly given in [45], but from Eq 3 in [45] we know that $\frac{RT}{F} = 26.64 \cdot 10^{-3}$V. By using the values of $R$ and $F$ listed in Table 3, we get an absolute temperature of 309.14 K, corresponding to a body temperature of 36˚C.

At each time step, $\phi$ in all four compartments was derived algebraically:

$$\phi_{de} = 0, \tag{63}$$

$$\phi_{di} = Q_{di}/(c_m A_d) \tag{64}$$

$$\phi_{se} = \left(\phi_{di} - \frac{\Delta x}{\sigma_i} \cdot i_{diff,i} - \frac{A_e \Delta x}{A_i \sigma_i} \cdot i_{diff,e} - \frac{Q_{si}}{c_m A_s}\right) \bigg/ \left(1 + \frac{A_e \sigma_e}{A_i \sigma_i}\right), \tag{65}$$

$$\phi_{si} = \frac{Q_{si}}{c_m A_s} + \phi_{se}. \tag{66}$$

The total membrane flux densities were as follows:

$$j_{Na,sm} = j_{Na} + j_{Na,leak} + 3j_{pump} + j_{nkcc1} - 2j_{Ca-dec}, \tag{67}$$

$$j_{K,sm} = j_{K-DR} + j_{K,leak} - 2j_{pump} + j_{nkcc1} + j_{kcc2}, \tag{68}$$

$$j_{Cl,sm} = j_{Cl,leak} + 2j_{nkcc1} + j_{kcc2}, \tag{69}$$

$$j_{Ca,sm} = j_{Ca-dec}, \tag{70}$$

$$j_{Na,dm} = j_{Na,leak} + 3j_{pump} + j_{nkcc1} - 2j_{Ca-dec}, \tag{71}$$

$$j_{K,dm} = j_{K-AHP} + j_{K-C} + j_{K,leak} - 2j_{pump} + j_{nkcc1} + j_{kcc2}, \tag{72}$$

$$j_{Cl,dm} = j_{Cl,leak} + 2j_{nkcc1} + j_{kcc2}, \tag{73}$$

$$j_{Ca,dm} = j_{Ca} + j_{Ca-dec}. \tag{74}$$

Fig 2 summarizes the model. The parameters involved in this model and their values used in this study are listed in Tables 1–4.

## Original Pinsky-Rinzel model

We implemented the original Pinsky-Rinzel equations from Box 8.1 in [8]. The reversal potential of the leak current, not specified in [8], was set to -68 mV to ensure a resting potential close to that of the edPR model. We also used this as the initial potentials, that is, $\phi_{sm,0} =$

**Table 4. Membrane parameters.**

| Parameter | Value | Reference |
|---|---|---|
| $c_m$ | $3 \cdot 10^{-2}$ F/m$^2$ | [3, 8] |
| $\overline{g}_{Na,leak}$ | 0.247 S/m$^2$ | [45] |
| $\overline{g}_{K,leak}$ | 0.5 S/m$^2$ | [45] |
| $\overline{g}_{Cl,leak}$ | 1.0 S/m$^2$ | [45] |
| $\overline{g}_{Na}$ | 300 S/m$^2$ | [3, 8] |
| $\overline{g}_{DR}$ | 150 S/m$^2$ | [3, 8] |
| $\overline{g}_{Ca}$ | 118 S/m$^2$ | |
| $\overline{g}_{AHP}$ | 8 S/m$^2$ | [3, 8] |
| $\overline{g}_C$ | 150 S/m$^2$ | [3, 8] |
| $\rho$ | $1.87 \cdot 10^{-6}$ mol/(m$^2$s) | [45]* |
| $U_{kcc2}$ | $7.0 \cdot 10^{-7}$ mol/(m$^2$s) | [45]* |
| $U_{nkcc1}$ | $2.33 \cdot 10^{-7}$ mol/(m$^2$s) | [45]* |
| $U_{Ca-dec}$ | 75 s$^{-1}$ | [3, 8] |

* We multiplied the original values from [45] by a conversion factor $\frac{7}{3} \cdot 10^{-6}$ m to convert the units from mM/s to mol/m$^2$s. The conversion factor equals the initial inverse surface area to volume ratio from [45].

−68mV and $\phi_{dm,0} = -68$mV. The other initial conditions were $n_0 = 0.001$, $h_0 = 0.999$, $s_0 = 0.009$, $c_0 = 0.007$, $q_0 = 0.01$, and $[Ca^{2+}]_0 = 0.2$, same as in [3].

## Simulations

**Parameterizations.** The parameters listed in Tables 1–4 were used in all the simulations of the electrodiffusive Pinsky-Rinzel (edPR) model. We tuned the Ca$^{2+}$ conductance $\overline{g}_{Ca}$ manually to obtain comparable spike shapes between the edPR model and the original PR model, as well as the fraction of free Ca$^{2+}$ inside the cell, and the coupling strength between the soma and the dendrite.

In the edPR model, the coupling strength between the soma and dendrite was proportional to the ratio $A_i/\Delta x$, and all model outputs depended on this ratio, and not on $A_i$ or $\Delta x$ in isolation. By choice, we adjusted the coupling strength by varying $A_i = \alpha A_m$ through adjusting the parameter $\alpha$. We could have obtained the equivalent effect by varying $\Delta x$ instead. The default value of $\alpha$ was set to 2. All simulations were run using this value, except in Fig 5C where $\alpha$ was set to 0.43.

In the original PR model, the coupling strength between the soma and dendrite was represented by a coupling conductance $g_c$, which had a default value of 10.5mS/cm$^2$. In Fig 5A, $g_c$ was set to 2.26mS/cm$^2$.

**Initial conditions.** The initial conditions for the edPR model were obtained through a two-step procedure. In the first step, we specified a set of pre-calibration initial values: We set (i) the initial membrane potential, $\phi_{m,0}$, to -68 mV, (ii) the concentrations to the pre-calibrated values in Table 5, and (iii) the gating variables (Table 5) to the same initial values as in [3]. Based on the initial concentration values, we also defined (iv) a set of static intracellular and extracellular residual charges, representing negatively charged macromolecules present in real neurons. We represented these as constant concentrations ($[X^-]_{i,0}$ and $[X^-]_{e,0}$) of anions with charge number $z_X = -1$ (assuming this to be the mean charge number of the real macromolecules) and diffusion constants $D_X = 0$ (assuming immobility). The residual charges were introduced to ensure consistency between the initial membrane potential and the charge density

**Table 5. Initial conditions.**

| Variables | Pre-calibrated | Post-calibrated[1] |
|---|---|---|
| $\phi_{m,0}$ [†] | -68 mV | -67.7 mV |
| $[Na^+]_{i,0}$ | 15 mM | 16.9 mM |
| $[Na^+]_{e,0}$ | 145 mM | 141.2 mM |
| $[K^+]_{i,0}$ | 140 mM | 139.5 mM |
| $[K^+]_{e,0}$ | 5 mM | 5.9 mM |
| $[Cl^-]_{i,0}$ | 4 mM | 5.4 mM |
| $[Cl^-]_{e,0}$ | 110 mM | 107.1 mM |
| $[Ca^{2+}]_{i,0}$ | 0.01 mM[*] | 0.01 mM[*] |
| $[Ca^{2+}]_{e,0}$ | 1.1 mM | 1.1 mM |
| $[X^-]_{i,0}$ [‡] | 151.0 mM | 151.0 mM |
| $[X^-]_{e,0}$ [‡] | 42.2 mM | 42.2 mM |
| $n_0$ | 0.001 | 0.0003 |
| $h_0$ | 0.999 | 0.999 |
| $s_0$ | 0.009 | 0.007 |
| $c_0$ | 0.007 | 0.005 |
| $q_0$ | 0.010 | 0.011 |
| $z_0$ | 1.0 | 1.0 |

[1] Preciser values (with more decimals included) were read to/from file and used in the simulations. (Available at https://github.com/CINPLA/EDPRmodel_analysis).

[†] $\phi_m$ is not an independent state variable, but at each time point an algebraic function of ion concentrations, as computed through the KNP formalism.

[*] Only 1% of the total intracellular $Ca^{2+}$, that is, a 100 nM, was assumed to be free (unbuffered).

[‡] Not state variables, but constants, derived (initially) from Eqs 75 and 76.

associated with the initial ion concentrations:

$$[X^-]_{i,0} = z_{Na}[Na^+]_{i,0} + z_K[K^+]_{i,0} + z_{Cl}[Cl^-]_{i,0} + z_{Ca}[Ca^{2+}]_{i,0} - \phi_{m,0}\frac{c_m A_m}{V_i F}, \tag{75}$$

$$[X^-]_{e,0} = z_{Na}[Na^+]_{e,0} + z_K[K^+]_{e,0} + z_{Cl}[Cl^-]_{e,0} + z_{Ca}[Ca^{2+}]_{e,0} + \phi_{m,0}\frac{c_m A_m}{V_e F}. \tag{76}$$

In the next step, we calibrated the model by running it for 1800 s to obtain the post-calibrated values of all the state variables (Table 5). These post-calibrated values were written to file and used as initial conditions in all simulations shown throughout this paper. For technical reasons, we did not read the constant residual concentrations, $[X^-]_{i,0}$ and $[X^-]_{e,0}$, to/from file, but re-calculated them from Eqs 75 and 76 in the beginning of each simulation to minimize rounding errors and ensure strict electroneutrality. While the pre-calibrated initial conditions were identical in the somatic and dendritic compartment, the post-calibration values were not strictly identical, but identical up to the decimal place included in Table 5. Hence, the indicated values apply for both the soma and dendrite compartments. The post-calibrated values of the ion concentrations gave us the following reversal potentials: $E_{Na}$ = 57mV, $E_K$ = −84mV, $E_{Cl}$ = −79mV, and $E_{Ca}$ = 124mV.

The pre-calibration values for the ion concentrations were taken from Table 2.1 in [85], which lists the ranges of typical intra- and extracellular concentrations in mammalian neurons.

From the ranges given in this table, we selected values that made the edPR model (throughout the calibration period) reside close to the selected initial membrane potential of -68 mV, a typical value found for CA3 pyramidal cells in adult rats [86].

**Stimulus current.** We stimulated the cell by injecting a $K^+$ current $i_{stim}$ into the soma. Previous computational modeling of a cardiac cell has shown that stimulus with $K^+$ causes the least physiological disruption [33]. To ensure ion conservation, we removed the same amount of $K^+$ ions from the corresponding extracellular compartment:

$$\frac{d[K^+]_{si}}{dt} + = \frac{i_{stim}}{Fz_K V_{si}}, \tag{77}$$

$$\frac{d[K^+]_{se}}{dt} - = \frac{i_{stim}}{Fz_K V_{se}}. \tag{78}$$

**Analysis.** Fig 9: To calculate the accumulative transport of ion species k in the intracellular solution (from time zero to $t$) due to diffusion, we integrated $A_i N_A j_{k,diff, i}$ from time zero to t, where $N_A$ is the Avogadro constant. Similarly, we integrated $A_e N_A j_{k,diff,e}$ to calculate the accumulative transport of ions in the extracellular solution due to diffusion. We did the same calculations with $j_{k,drift}$ to study the accumulative transport of ions due to drift. When knowing the accumulative transport of each ion species, $k_{akkum}$, we calculated the total transport of $e^+$ from their weighted sum:

$$e^+_{akkum} = z_{Na} Na^+_{akkum} + z_K K^+_{akkum} + z_{Cl} Cl^-_{akkum} + z_{Ca} Ca^{2+}_{akkum}. \tag{79}$$

Fig 10: To calculate $\phi_{VC,e}$ and $\phi_{diff,e}$, we looked at the extracellular axial current as it is given in the KNP formalism:

$$i_e = i_{diff,e} + i_{field,e} = i_{diff,e} + \sigma_e \frac{\phi_{se}}{\Delta x}, \tag{80}$$

where the last equality follows when we insert Eq 18 for the extracellular field-driven current density $i_{field,e}$, and use that $\phi_{de} = 0$. As in [32], we may split $\phi_{se}$ into two components:

$$\phi_{se} = \phi_{VC,se} + \phi_{diff,se}, \tag{81}$$

where $\phi_{VC,se}$ is the potential as it would be predicted from standard volume conductor (VC) theory [20, 21], and $\phi_{diff,se}$ is the additional contribution from diffusion [32]. With this, Eq 80 can be written:

$$i_e = i_{diff,e} + \sigma_e \frac{\phi_{VC,se}}{\Delta x} + \sigma_e \frac{\phi_{diff,se}}{\Delta x}. \tag{82}$$

We may split this into two equations if we recognize that

$$i_e = \sigma_e \frac{\phi_{VC,se}}{\Delta x}, \tag{83}$$

is the standard formula used in VC theory, which is based on the assumption that the extracellular current is exclusively due to a drop in the extracellular VC-potential $\phi_{VC,se}$. The

remainder of Eq 82 then leaves us with

$$i_{\text{diff,e}} = -\sigma_{\text{e}} \frac{\phi_{\text{diff,se}}}{\Delta x}. \tag{84}$$

Since we already knew $i_{\text{e}}$ and $i_{\text{diff,e}}$ from simulations with the KNP framework, we used Eqs 83 and 84 to calculate $\phi_{\text{VC,se}}$ and $\phi_{\text{diff,se}}$ separately.

**Numerical implementation.** We implemented the differential equations in Python 3.6 and solved them using the `solve_ivp` function from SciPy. We used its default Runge-Kutta method of order 5(4), and set the maximal allowed step size to $10^{-4}$. The code is made available at https://github.com/CINPLA/EDPRmodel and https://github.com/CINPLA/EDPRmodel_analysis.

## Author Contributions

**Conceptualization:** Marte J. Sætra, Gaute T. Einevoll, Geir Halnes.

**Formal analysis:** Marte J. Sætra, Geir Halnes.

**Funding acquisition:** Gaute T. Einevoll.

**Investigation:** Marte J. Sætra, Geir Halnes.

**Methodology:** Marte J. Sætra, Geir Halnes.

**Project administration:** Gaute T. Einevoll, Geir Halnes.

**Software:** Marte J. Sætra.

**Supervision:** Geir Halnes.

**Validation:** Marte J. Sætra, Geir Halnes.

**Visualization:** Marte J. Sætra.

**Writing – original draft:** Marte J. Sætra, Geir Halnes.

**Writing – review & editing:** Marte J. Sætra, Gaute T. Einevoll, Geir Halnes.

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
