## [Decision Letter · Decision Letter 0]

25 Feb 2020

Dear Dr. Halnes,

Thank you very much for submitting your manuscript "An electrodiffusive, ion conserving Pinsky-Rinzel model with homeostatic mechanisms" for consideration at PLOS Computational Biology. As with all papers reviewed by the journal, your manuscript was reviewed by members of the editorial board and by several independent reviewers. The reviewers appreciated the attention to an important topic. Based on the reviews, we are likely to accept this manuscript for publication, providing that you modify the manuscript according to the review recommendations.

Sincerely,

William W Lytton, MD

Guest Editor

PLOS Computational Biology

Kim Blackwell

Deputy Editor

PLOS Computational Biology

[LINK]

Reviewer's Responses to Questions

**Comments to the Authors:**

Reviewer #1: The paper applies the Kirchoff-Nernst-Plank framework (a method for modeling electrodiffusion) to a reduced CA3 cell model. The original two compartment model is augmented with two additional extracellular compartments, modeling of both the intracellular and extracellular potentials, electrodiffusion of K+, Na+, Cl- and Ca2+ and several homoeostatic mechanisms. The model qualitatively matches the original, but unlike the simple model they started from the electrodiffusive model can be driven into depolarization block by excessive stimulation or depolarize when the homoeostatic mechanisms are impaired. This novel and innovative approach will be useful for the studying the mechanisms underling pathological conditions like ischemia and epilepsy, that can lead to spreading depolarization.

The sodium calcium exchanges takes in 3 sodium for every calcium ion at the cost of 1 ATP. Why do you have it exchanging 2 sodium ions instead? Eq 67 & 71.

The tortuosity is mentioned in the methods section, the value results from an increase in path length as ions diffusion around obstacles. Does this make sense for your model? If so, why haven't you also included the free volume fraction?

This seems like it could be an issue for interpreting the models boundary conditions as equivalent to period boundary conditions for a synchronized population, as the population would have a relatively large extracellular space.

Figure 4: Are the ion concentrations as stable as the somatic membrane potential?

Do all the simulations remain stable over the whole +/-15% parameter change or are some slowly depolarizing, like a less extreme 8A? (or hyperpolarizing).

Figure 6 and 7: Plotting changes freely diffusion calcium on the same scale as sodium is not useful. Why not plot the states as you did for figure 8?

Figure 8:

Why do you turn off all of the homoeostatic mechanisms, rather than just those two that depend on ATP?

G] Why does the intracellular dendritic calcium differ so much from the intracellular somatic calcium?

Initial conditions: How did you adjust the ion concentrations to arrive at stable initial condition? Would the model be unstable if you used initial with concentration from the literature? e.g. Ions in the Brain, Somjen. GG, 2002.

Also, why didn't you use the post calibration states as your initial conditions? What do the first 15s of the simulations look like?

Does this mean the initial (post-calibration) conditions for the sensitivity analysis were different for each simulations?

Minor issues/typos;

Line 392: ICRP model -- only mentioned here.

Figure 10: figure legend is not the same as the caption.

Table 1 caption: Two As, both intracellular volumes don't correspond to a sphere with radius 7um.

Reviewer #2: This paper proposed an electrodiffusive Pinsky-Rinzel (edPR) model, which is “a minimal neuronal model” with two neuronal compartments (a soma and a dendrite), plus two extracellular compartments (outside soma and outside dendrite), by adding homeostatic mechanisms and ion concentration dynamics. The edPR model doesnot only reproduce the membrane potential dynamics of the PR model, but also accounts for changes in neuronal firing properties due to deviations from baseline ion concentrations. They expect the model to be important because it opens for more detailed mechanistic studies of the pathological conditions associated with large changes in ion concentrations such as epilepsy and spreading depression.

Overall, the article is well organized and presented. I have some minor comments here.

1. The statement of “the first multicompartmental neuron model that in a biophysically consistent way does account for the effects of ion concentration variations ” is too strong. For example, Kager et al 2000 is a morphologically explicit multicompartmental model with ion concentration variations.

2. What is the advantage of using the two-compartmental model compared with one-compartment model or morphologically explicit model?

3. The model has two neuronal compartment compartments (a soma and a dendrite), it is a little confused how did the authors investigate the ionic diffusion along the axons in the statement of “investigate the consequences of neglecting the effect of ionic diffusion (along dendrites and axons) on the electrical potential”.

4. Table 3, T is body temperature at 36C (309.15K).

**Have all data underlying the figures and results presented in the manuscript been provided?**

Reviewer #1: Yes

Reviewer #2: None

PLOS authors have the option to publish the peer review history of their article (what does this mean?). If published, this will include your full peer review and any attached files.

Reviewer #1: No

Reviewer #2: No
---

## [Decision Letter · Decision Letter 1]

7 Apr 2020

Dear Dr. Halnes,

We are pleased to inform you that your manuscript 'An electrodiffusive, ion conserving Pinsky-Rinzel model with homeostatic mechanisms' has been provisionally accepted for publication in PLOS Computational Biology.

Best regards,

William W Lytton, MD

Guest Editor

PLOS Computational Biology

Kim Blackwell

Deputy Editor

PLOS Computational Biology

Reviewer's Responses to Questions

**Comments to the Authors:**

Reviewer #1: Thank you for addressing and answering all my issues and questions.

**Have all data underlying the figures and results presented in the manuscript been provided?**

Reviewer #1: Yes

PLOS authors have the option to publish the peer review history of their article (what does this mean?). If published, this will include your full peer review and any attached files.

Reviewer #1: No

---

## [Editor Report · Acceptance letter]

20 Apr 2020

PCOMPBIOL-D-20-00066R1 

An electrodiffusive, ion conserving Pinsky-Rinzel model with homeostatic mechanisms

Dear Dr Halnes,

I am pleased to inform you that your manuscript has been formally accepted for publication in PLOS Computational Biology. Your manuscript is now with our production department and you will be notified of the publication date in due course.

With kind regards,

Sarah Hammond
